# Monitoring compliance with fuel sulfur content regulations of sailing ships by unmanned aerial vehicle (UAV) measurements of ship emissions in open water

Fan Zhou[1,2], Liwei Hou[3], Rui Zhong[1,2], Wei Chen[4], Xunpeng Ni[4], Shengda Pan[1,2], Ming Zhao[1,2,5], Bowen An[1,2]

[1]College of Information Engineering, Shanghai Maritime University, Shanghai 201306, China
[2]Shanghai Engineering Research Center of Ship Exhaust Intelligent Monitoring, Shanghai 201306, China
[3]College of Ocean Science and Engineering, Shanghai Maritime University, Shanghai 201306, China
[4]Pudong Maritime Safety Administration of the People's Republic of China, Shanghai 200137, China
[5]Key Laboratory of Intelligent Infrared Perception, Chinese Academy of Sciences, Shanghai 200083, China

*Correspondence to*: Fan Zhou (fanzhou_cv@163.com)

**Abstract.** Due to technical and cost limitations, the monitoring of emissions from ships sailing in open water within the ship emission control areas (ECAs) is relatively rare. The present study adopts a monitoring method involving an unmanned aerial vehicle (UAV) that takes off from a patrol boat to measure the concentrations of $SO_2$ and $CO_2$ within the plumes of sailing ships. Our method aims to provide a low-cost, remote approach for estimating the fuel sulfur content (FSC) of sailing ships in open water, which overcomes the limitations of ground-based and small aircraft methods. The selected monitoring area was the Yangtze River estuary, a domestic ECA with an FSC limit of 0.5% (m/m) implemented by the Chinese government. A total of 27 sailing ships were monitored, 12 of which were found to have an FSC of > 0.5% (m/m). Moreover, the FSCs of the sailing ships were found to be higher than those of berthing ships in the study area. Based upon the online monitoring results, four of the monitored ships were intercepted by the maritime law enforcement, and fuel samples were collected and analyzed in a laboratory; the results confirmed that all four FSCs were > 0.5% (m/m). Among them, one offending ship was tracked down on July 15, 2019, which was the first time that a sailing ship had been caught for having failed the FSC regulations in China. Overall, the present study provides scientific support for evaluating the effectiveness of ECA policies, and recommends that emissions from sailing ships should be monitored more often in the open water in the future.

## 1. Introduction

With the rapid development of the shipping industry (UNCTAD, 2017) over the past decades, air pollution caused by ship emissions has received an increasing amount of attention (Eyring et al., 2010; Wan et al., 2016). The pollutant gases emitted by ships not only affect the global climate (Huebert, 1999; Corbet, 2016), but also local air quality and can harm people's health. (Yang et al., 2016; Wang et al., 2019). Shipping accounted for 15%, 13%, and 3% of the annual global anthropogenic emissions of NOx, SOx, and $CO_2$ from 2007 to 2012, respectively (Smith et al., 2014). In Europe, estimated ship emissions

were responsible for 3.0 million tons of NOx, 1.2 million tons of SOx, and 0.2 million tons of fine particulate matter (PM2.5) in 2011 (Jalkanen et al., 2016). In East Asia, shipping emissions accounted for 16% of global shipping $CO_2$ in 2013, whereas they only accounted for 4–7% during 2002–2005 (Liu et al., 2016).

To reduce the negative impacts of ship emissions, the International Maritime Organization (IMO) regulates emissions through the International Convention for the Prevention of Pollution from Ships and its Annex VI (MARPOL, 1997). The air-pollution limits for shipping were adopted in 1997, but only came into force in 2005. The global cap for the fuel sulfur content (FSC) of seagoing ships was set at 3.5% (m/m) in 2012, and was reduced to 0.5% (m/m) in 2020. To date, four emission control areas (ECAs) (the Baltic Sea, the North Sea, the United States Caribbean, and the North American and United States Caribbean Sea) have been set up, and the corresponding FSC limit for seagoing ships in these areas was set at 0.1% (m/m) in 2015 (IMO, 2017).

The IMO has not yet set up ECAs in East Asia, which includes the world's ten largest container ports, for example, Shanghai, Ningbo-Zhoushan, and Shenzhen ports. To limit the air pollution caused by ship emissions, the Chinese government established three domestic emission control areas (DECAs) in 2015: the Yangtze River delta, the Pearl River delta, and the Bohai Sea. DECAs was expanded to cover a wider area since 2020, and include most of the coastal ports, the Yangtze River main line, and the Xijiang River main line. The FSC limit for sailing and berthing ships in the DECAs has been set at 0.5% (m/m) since January 1, 2019.

A key problem regarding the implementation of the policy of the ECAs is the question of how to enforce the FSC of ships. Several studies have suggested estimating FSC by measuring ship plumes (Berg et al., 2012; Balzani Lööv et al., 2014). At present, the main method to monitor the emissions of surrounding ships is to place monitoring equipment either on the wharf, shore, port area, or bridge (i.e., ground-based methods) (Alföldy et al., 2013; Pirjola et al., 2013; Beecken et al., 2015; Kattner et al., 2015; Mellqvist et al., 2017a; Cheng et al., 2019; Zhang et al., 2019). Although ground-based methods can provide continuous monitoring, the results obtained depend on the wind speed, wind direction, and the relative position of a ship to the monitoring equipment. Additionally, the boundaries of the ECAs that are designated by the IMO are 200 nautical miles from the coast (Viana et al., 2015); hence, ground-based methods are not able to monitor the fuel that is used on the open sea in ECAs because sailing ships are too far from the shore or bridges.

Therefore, some researchers have used sensors that are carried by small aircrafts to monitor navigating ships within ECAs (Berg et al., 2012; Beecken et al., 2014). However, because this kind of monitoring method is costly, the monitoring of navigating ships is relatively rare. Beecken et al. (2015) observed 434 plumes during ground-based measurements and 32 plumes from a helicopter. Balzani Lööv et al. (2014) took 475 measurements using "sniffing" instruments from ground-based measurements, whereas only 25 measurements were obtained using this method from mobile platforms. In the study undertaken by Mellqvist et al. (2017b), 114 individual ships were measured effectively during 27 flight hours at a cost of approximately 470 Euro per ship, which was for the aircraft cost and did not included the ferry, operator, or instrument rental costs. Therefore, the high cost of flying precludes extensive monitoring of ship emissions.

As a result of the aforementioned factors, there is less monitoring of ships on the open sea in ECAs. This is despite the fact that numerous studies (Pirjola et al., 2014; Kattner et al., 2015; Zhang et al., 2019) have shown that the FSC of ships were significantly reduced by the implementation of the ECA policy. However, most of these studies did not involve the monitoring of ships on the open water, which could lead to non-representative assessments for the implementation of policies. At the same time, the lack of open sea monitoring results in a blind area for maritime enforcement and is not conducive to the implementation of ship ECA policy by maritime authority. The present study used an unmanned aerial vehicle (UAV) to monitor the FSC of sailing ships on the open sea in the Yangtze River estuary DECA. The method proposed in this study can be used to monitor ship emissions at a comparatively low cost to understand the FSCs of sailing ships in open waters. Although the cost of using patrol boats is not negligible, it is more convenient and cheaper for maritime authorities than using small aircraft.

## 2 Experimental methods

The research undertaken in the present study forms part of the project "Monitoring and inspecting ship exhaust emissions in the Shanghai free-trade zone" (MISEE). In this project, a UAV system was designed and developed, and mainly included a pod for measuring the exhaust gas from ships and a UAV to carry the pod. In previous research (Zhou et al., 2019), the plumes of 23 berthing ships were measured using the first-generation pod, and the deviation of the estimated FSC was $< 0.03\%$ (m/m) for an FSC of between 0.035% (m/m) and 0.24% (m/m).

In the present monitoring for sailing ships, we developed the second-generation pod by optimizing the structure and layout of the first-generation pod to achieve a lighter weight and smaller volume. A short overview of the instrumentation is provided in Section 2.1. We measured the plumes of 11 berthing ships to verify the accuracy of the second-generation pod, and the plumes of 27 sailing ships to estimate the FSC.

## 2.1 Instrumentation

The instrumentation that was used for monitoring the FSC of sailing ships is shown in Figure 1. The UAV was a MATRICE 600 PRO (SZ DJI Technology Co., Ltd., Shenzhen, China). This type of UAV cannot be used on rainy days or when the wind speed is higher than 8 m/s. The white box installed underneath the UAV in Fig. 1 is the aforementioned second-generation pod for measuring the exhaust gas. When the UAV approaches a ship's plume, the gas pump in the pod draws air using the gas probe. The water vapor, particles, and soot in the gas are subsequently removed by a hose filter valve. The sensors detect the gas and measurement information is sent out by communication modules. The pod has dimensions of 20 cm $\times$ 12 cm $\times$ 9 cm and weighs 900 g.

The sensors used were able to measure both $SO_2$ and $CO_2$, and were purchased from Shenzhen Singoan Electronic Technology Co., Ltd., China. The $SO_2$ sensor is based on the electrochemical method, and has a measuring range of 0–10 ppm, an accuracy of $\pm 3\%$ (0.3 ppm), and a response time ($T_{90}$) of $\leq 30$ s. The $CO_2$ sensor is based on the non-dispersive infrared analyzer method,

and has a measuring range of 0–10000 ppm, an accuracy of $\pm$ 3% (300 ppm), and a $T_{90}$ of $\leq$ 30 s. The $T_{90}$ represents the time taken to reach 90% of the stable response following a full range change in the sample concentration. These sensor characteristics were provided by the instrument manufacturer and were ensured to be within the tolerances by calibration. The zero and full scales are usually calibrated by a standard mixed gas when the equipment is used on a daily basis. The major parameters of the UAV system are listed in Table 1.

## 2.2 Monitoring region

As illustrated in Fig. 2, the monitoring region was the channel of the Yangtze River estuary, near the Waigaoqiao port area to the north of Shanghai. The Yangtse River is the first (third) longest river in China (the world). Shanghai is one of the most prosperous cities in the world, and at the end of 2017 that city had a permanent resident population of approximately 24 million people (Shanghai Municipal Bureau of Statistics, 2017). The Waigaoqiao port area is only 20 km away from the city center, and the air pollution caused by ship emissions directly affects the urban air environment and the health of residents (Wang et al., 2019; Feng et al., 2019). The experimental area of the MISEE project is mainly within the Waigaoqiao port and the Yangtze River estuary.

## 2.3 Measurement method

During the experiment, the operator took a patrol boat to the channel and then selected a target ship at random. After identifying the target ship for monitoring, the patrol boat would accelerate to a distance to the left or right ahead of the vessel. The patrol boat would then stop and the UAV was operated to takeoff from its deck, and would then fly towards the plume of the target ship and measure the concentrations of $SO_2$ and $CO_2$ in the plume (Fig. 3). The distance between the patrol boat and the target ship was a few hundred meters.

During the measurements, the operator adjusted the position of the UAV to ensure that it was in the ship's plume. Real-time measurements of $SO_2$ and $CO_2$ were made such that the pod could effectively detect the plume. Generally, it was necessary for the UAV to follow the ship's funnel mouth for approximately 5 minutes, as illustrated in Fig. 4. The target ship continued to move during the measurements; hence, it was followed by the patrol boat in order to avoid the UAV moving too far away from the operator. When the operator was sure that valid data had been collected, the patrol boat stopped and the UAV returned and landed back on the deck of the patrol boat.

## 2.4 Calculation

The FSC in this study was obtained directly by sampling the gas concentrations in the ship plumes using the UAV. The enhancements of $SO_2$ and $CO_2$ in measurements that were affected by exhaust gases were calculated, and the ratio of these $SO_2$ and $CO_2$ peaks was used to calculate the FSC (Eqs. 1 and 2). This method has been widely used to calculate the FSC in related studies (Alföldy et al., 2013; Pirjola et al., 2014; Balzani Lööv et al., 2014; Beecken et al., 2014; Beecken et al., 2015; Kattner et al., 2015; Zhou et al., 2019). In the calculation, the molecular weights of carbon and sulfur are 12 g mol$^{-1}$ and 32 g

mol$^{-1}$, respectively, and the carbon mass percent in the fuel is 87 ± 1.5% (Cooper et al., 2003). By assuming that 100% of the carbon content of the fuel is emitted as $CO_2$, and sulfur is emitted as $SO_2$ and other forms, the FSC mass percent can be determined using Eq. (1):

$$FSC[\%] = \frac{S[kg]}{fuel[kg]} = \frac{SO_2[ppm] \cdot A(S)}{CO_2[ppm] \cdot A(C)} \cdot 87[\%] + R = 0.232 \frac{\int (SO_{2,peak} - SO_{2,bkg})dt[ppb]}{\int (CO_{2,peak} - CO_{2,bkg})dt[ppm]}[\%] + R = \frac{1}{20}EF[g_{SO_2}/kg_{fuel}] + R, \quad (1)$$

where R represents the sulfur content that is emitted in forms other than $SO_2$ because preliminary studies have shown that 1–19% of the sulfur in fuel is emitted in other forms, possibly $SO_3$ or $SO_4$ (Schlager et al., 2006; Alföldy et al., 2013; Balzani Lööv et al., 2014). EF is the emission factor and bkg is the abbreviation of background. In Eq. (1), if the sensors measuring $SO_2$ and $CO_2$ have approximately the same response time and can be set to be synchronized, the peak concentrations of $SO_2$ and $CO_2$ can be used to calculate the FSC; otherwise, integrals need to be used. In our research, the sampling rate of the $SO_2$

and $CO_2$ sensors was 1 s, and integrals were used because the two sensors could not be completely synchronized.

The continuous measurement data for two typical plumes (2019-4-15B and 2019-3-29A) are exhibited in Fig. 5. The data for plume 2019-4-15B (Fig. 5a) were considered to be of a "good" quality, whereas those for plume 2019-3-29A (Fig. 5c) were considered to be of a "poor" quality. Data were determined to be of a good-quality when obvious, easily distinguished peak values were observed, whereas less obvious peaks that still corresponded to a result were considered as poor-quality data.

Meanwhile, the correlation between the $SO_2$ and $CO_2$ time series is a key factor in judging quality. Assuming that the gas is completely mixed, the variation trend of the $SO_2$ and $CO_2$ measurements should be the same (although there may be some deviation because the corresponding time of the sensor was not consistent) and can be identified in the peak area.

The selection of peak values leads to uncertainty because when the area ratio is selected for the calculation, the starting and ending time points of the area are still associated with substantial uncertainty. Figure 5b and 5d depict the average

concentrations of the $SO_2$ and $CO_2$ measurements (in Fig. 5a and 5c, respectively) for 10 s periods. The peak value of each average concentration was selected for the calculation. This process is equivalent to selecting the area ratio of $SO_2$ to $CO_2$ within 10 s for the calculation, as shown in Eq. (2).

$$FSC[\%] = 0.232 \frac{\int (SO_{2,peak} - SO_{2,bkg})dt[ppb]/10}{\int (CO_{2,peak} - CO_{2,bkg})dt[ppm]/10}[\%] + R \approx 0.232 \frac{AVG(SO_{2,peak}) - AVG(SO_{2,bkg})}{AVG(CO_{2,peak}) - AVG(CO_{2,bkg})}[\%], \quad (2)$$

where $AVG(\cdot)$ is the calculated function for the average measurement value within 10 s; hence, the data in this study are the

average values of measurements in 10 s. When the UAV took off from the patrol boat and flew high into the air, the $SO_2$ and $CO_2$ concentrations were relatively low. The background values were obtained at this stage as the minimum $SO_2$ and $CO_2$ concentrations. As the UAV flew into the plume, the measured concentrations of $SO_2$ and $CO_2$ increased. The obvious, stable maximum values in the observations of the average measurement values should be selected as the peak values. It can be seen that using the average values of measurements within 10 s makes it easier to select the peak values, especially with respect to

poor-quality data. However, as there can still be several options for peak values, the data treatment methods reported by Zhou et al. (2019) were incorporated in this study to select the most appropriate peak values. In Fig. 5b, the time point of selected peak values is at 10:19:11. The measurement values from 10:19:57 to 10:20:15 were not used because the $CO_2$ concentration

covered the full range. In Fig. 5d, the time point of the selected peak values is at 10:38:27. The measurement values from 10:39:57 to 10:41:41 were not used because we ruled out data exhibiting either dramatic changes or errors in continuous observations. The details for selecting the peak values are listed in Table 2.

## 2.5 Uncertainties

In previous research (Zhou et al., 2019), the main uncertainties of UAV measurements were summarized as sensor uncertainty, measurement uncertainty, calculation uncertainty, and exhaust uncertainty. The instrument calibration method, UAV flight procedures, and data treatment methods were designed to reduce these uncertainties. However, some uncertainties remain, as discussed below.

To make it lightweight and convenient, the second-generation pod was only equipped with $SO_2$ and $CO_2$ sensors and a simple filter. We did not account for the interference that some factors might have caused, including that due to 1) the cross-sensitivity of the $SO_2$ sensor to $NO_2$, 2) the impact of a large temperature change in the exhaust plume, and 3) water vapor and/or particle contamination of the instruments.

The average gas concentration within 10 s was chosen for the FSC calculations; however, this does not mean that 9 s or 11 s could not have been selected. To demonstrate this, a comparison calculation was carried out using both 9 s and 11 s, which showed that these led to very little differences in the results. However, it is necessary to ensure that the gradient of the gas measurements is stable within the sampling time (the interval length of the integral). Moreover, the interval length cannot be too short (e.g., 2 s) or too long (e.g., 20 s). If the time is too short, it is difficult to determine whether the measurements are stable and undisturbed over time. Similarly, if the time is too long, it is also difficult to ensure that all of the measurements in the integral interval are stable and undisturbed. In addition, during the flight of the UAV in this study, the time available for measuring the plume was ~5 minutes. As both the ship and the UAV were moving at this time, it was virtually impossible to ensure that the UAV was flying consistently within the plume and obtaining stable measurements. Accordingly, 10 s is also a relatively appropriate value for the measurement process.

Nevertheless, there is also some uncertainty associated with choosing the peak values. After ruling out the peak values across the full range as well as those corresponding to dramatic changes, the global maximum values were selected as the peak values to calculate the FSC. The maximum values probably correspond to the measurements taken in the center of the ship's plume. At that location, the measurement values were relatively stable, and the probability of interference from other factors was lower. Furthermore, the higher the peak value is, the greater the proportion of exhaust gas is; hence, the impact from the incomplete mixing of the exhaust gas with clean air is relatively small.

In summary, the obvious and stable maximum values are selected as peak values to calculate the FSC. There are, of course, situations where multiple similar peaks can occur simultaneously. In this case, their calculated FSCs may be very similar, and the results obtained by the calculation of the highest peak should have high credibility, for instance, the measurements of plume 2019-4-15B. Meanwhile, the occurrence times of the peak $SO_2$ and $CO_2$ values sometimes have a small deviation that usually corresponds to a few seconds. This is due to two different sensor response times, which leads to three different options for

selecting the peak values: 1) the time point of the peak $SO_2$ value with the $CO_2$ value at the same time; 2) the time point of the peak $CO_2$ value with the $SO_2$ value at the same time; 3) the peak $SO_2$ and $CO_2$ values at different time points. Option 3 was selected in this research.

Additional uncertainties were encountered during our monitoring of sailing ships because the UAV was usually hundreds of meters away from the operator. The location of a plume depended primarily on the following three aspects. 1. The position of most plumes with black smoke could be identified through the operator's visual judgment. 2. The real-time image shot by the camera can be used to assist in finding the ship's funnel mouth. 3. In the measurement process, the real-time measured concentration sent to the receiving equipment gradually increased, thus indicating that the UAV was approaching the center of the plume. However, the operator occasionally faced difficulties in accurately determining the ship's plume, which led to failed measurements. We attempted to measure more than 40 ship plumes in open water; however, only 27 of them resulted in good- or poor-quality data, i.e., usable data.

The deviation of the estimated FSC value obtained by the first-generation pod was < 0.03% (m/m) for an FSC level ranging from 0.035% (m/m) to 0.24% (m/m) (Zhou et al., 2019). The second-generation pod was also verified on berthing ships by using this method at a similar FSC level and the accuracy was approximately the same (see Section 3.1). These verifications of the deviation were based on the FSC measurement of berthing ships, which did not exceed the Chinese DECA FSC limit of 0.5% (m/m). However, some of the sailing ships did exceed this limit. It should be noted that the deviations for different FSC levels were not the same. Based on previous studies, the deviation of the FSC obtained from high-sulfur plume should be greater, for example, Van Roy and Scheldeman (2016a, b) estimated relative uncertainties of 20% at a level of 1% (m/m) FSC and 50–100% at 0.1% (m/m) FSC. Therefore, the deviation of sailing ships may > 0.03% (m/m) when the FSC exceeds 0.5% (m/m). Nonetheless, our UAV system was still able to accurately detect an FSC that obviously exceeded 0.5% (m/m).

## 3. Results

### 3.1 Berthing ships

Before monitoring the sailing ships, we first monitored 11 berthing ships between March and April 2019 in the Waigaoqiao port to verify the accuracy of the second-generation pod. Whilst one person operated the UAV to monitor one of the plumes, two maritime law enforcement officers boarded the corresponding ship to collect a fuel sample. Both processes took approximately 10–20 min. The fuel samples, which are considered to represent the true FSC values, were then sent for chemical analysis in a laboratory. The estimated (UAV) and true FSC values are listed in Table 3 along with the identification number of each plume and the time and serial number. Table 3 shows that the deviation did not generally exceed 0.03% (m/m) for an FSC level of between 0.03% (m/m) and 0.22% (m/m) (except for plume 2019-3-22A and 2019-4-3B). Additionally, when the FSC of a target ship was low, for example, when light diesel fuel was used, the measured $SO_2$ concentrations were mostly zero. When this occurred, the FSC was generally < 0.02% (m/m), for example, as for plumes 2019-4-3A and 2019-4-12A.

### 3.2. Sailing ships and comparison with berthing ships

Between March and December 2019, effective monitoring of 27 sailing ships was undertaken using the UAV that took off from the patrol boat (Table 4). The FSC of 23 berthing ships measured by the first-generation monitoring equipment and the FSC of 11 berthing ships (Table 3) measured by the second-generation monitoring equipment in this study were taken as the FSC monitoring results for berthing ships. We compared the distribution of the FSCs of these 34 berthing ships with those of the 27 sailing ships. Figure 6 shows that the FSCs of the sailing ships were considerably higher than those of the berthing ships; the FSC of all 27 sailing ships exceeded 0.1% (m/m) and the FSC of 12 of these exceeded the Chinese DECA FCS limit of 0.5% (m/m), which included 5 exceedances of 1.5% (m/m). The uncertainty in the assessment is not small but the results so far, do not lead to optimism with respect to the FSC used by ships sailing in the area. The reason for this is that although berthing ships are sometimes boarded by maritime law enforcement officers for examination, an effective approach for monitoring the FSC of sailing ships in open water that leads to prosecution by China's maritime authorities has not existed prior to the present study.

According to the monitoring results, law enforcement officers of the Pudong maritime safety administration intercepted four sailing ships for which the UAV FSC results were of a good-quality and all exceeded 1.5% (m/m). The officers boarded these ships for inspection on July 15, August 14, August 20, and September 27, 2019, and took fuel samples, which were sent for chemical analysis in a laboratory. The FSC of all four fuels was also found to exceed 0.5% (m/m): 0.534% (m/m), 0.744% (m/m), 0.813% (m/m), and 1.991% (m/m) (in chronological order). The reason that three of these laboratory results did not exceed 1.5% (m/m) related to the fact that ships cannot stop immediately in the channel for inspection and have to sail to the anchorage point; when the officers boarded the ships to take samples they found the crew taking various measures to drain the high-sulfur fuel in the main engine fuel oil pipeline. This means that the chemical analysis results of the sampled fuels were obviously lower than those of the UAV monitoring. Nevertheless, the four inspections successfully confirmed that the FSC of the fuels exceeded the standard for sailing ships. The inspection on July 15, 2019, was the first time that a sailing ship's FSC failed to meet Chinese regulations, and this aroused wide concern in the shipping community.

### 4. Conclusions

In this research, we used a UAV that took off from a patrol boat to monitor emissions from sailing ships in open water. Of the 27 sailing ships that were successfully monitored, 12 were found to have an FSC that exceeded 0.5% (m/m) and 5 exceeded 1.5% (m/m). Based on the monitoring results, law enforcement officers of the Pudong maritime safety administration caught the first case of excessive FSC for a sailing ship and confirmed three other cases. Additionally, the UAV monitoring results demonstrated that the FSC values of sailing ships in the surrounding waters of Waigaoqiao port were higher than those determined for berthing ships in the port. Although the sample size was relatively small, observation of Fig. 6 suggests that the data are still convincing.

Although a global cap on the FSC in marine fuel was set at 3.5% (m/m) in 2012 following the IMO regulation, this was reduced to 0.5% (m/m) in 2020 and has already been implemented in China. According to our monitoring results, the current situation

for meeting the 0.5% (m/m) limit is not optimistic. Successful compliance with this regulation by ship owners involves many challenges. We conclude that there is a need for further monitoring data on sailing ships in open water to ascertain the degree of exceedance and work toward compliance.

In addition, there are still some improvements to be made to the UAV system. 4G transmission is the communication method for detecting information transmission; hence, in locations without a 4G signal (e.g., offshore), the receiving equipment cannot

obtain real-time measurement results. Potential solutions include setting-up small base-stations on patrol boats or using satellite transmission. Although carrying an infrared camera on the UAV would make it easier to find the plume, this would require to replace the camera in Fig. 1 with an infrared camera and establish new data communication.

**Data availability**

Please address requests for data sets and materials to Fan Zhou (fanzhou_cv@163.com).

**Author contribution**

FZ designed the study and authored the article. FZ and LH analyzed the experimental data. RZ, WC and XN contributed to the experiments. SP contributed to setting instruments. LH, MZ and BA provided constructive comments on this research.

**Competing interests**

The authors declare that they have no conflict of interest.

**Acknowledgements**

We thank Megan Anne for English language editing.

**Financial support.**

This research has been supported by the National Natural Science Foundation of China (grant No. 41701523), the Special

Development Fund for China (Shanghai) Pilot Free Trade Zone, the Special Foundation for Intelligent Manufacturing Industry of Shanghai Lin-Gang Area (grant no. ZN2017020325) and Open Project Program of Key Laboratory of Intelligent Infrared Perception, Chinese Academy of Sciences.

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

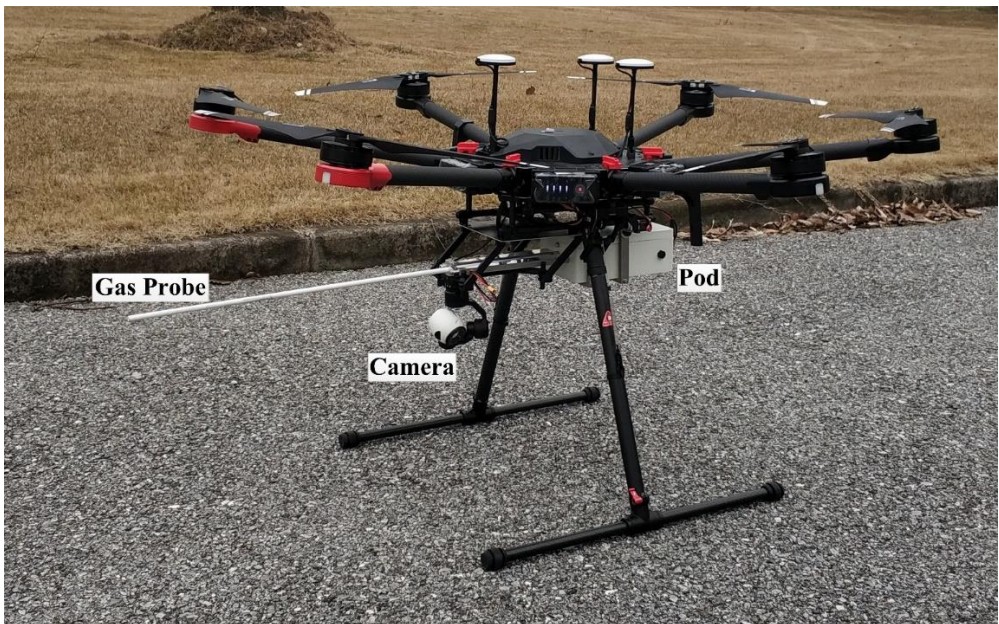

**Figure 1. Image of the UAV system. A gas probe, camera, and pod are installed under the unmanned aerial vehicle (UAV). The gas probe is used to collect the ship's exhaust gas, and the camera is used to assist in finding the ship's funnel mouth during flight. The pod is used to carry a gas pump, gas circuit, filter, small motor, sensors for $SO_2$ and $CO_2$, and communication modules.**

365

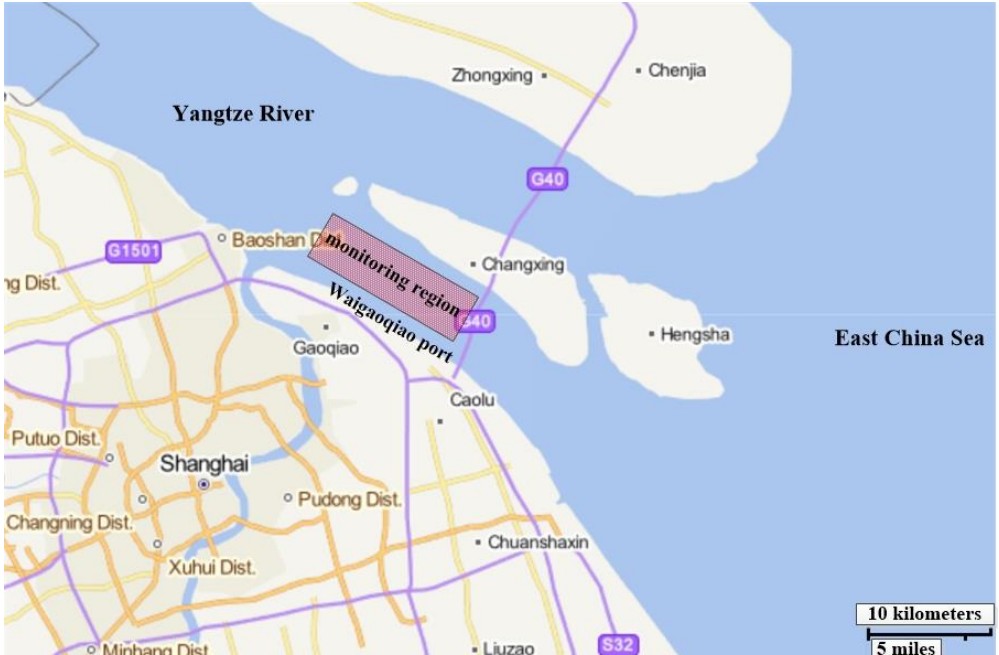

**Figure 2. Monitoring regions in the channel of Yangtze River estuary, which belong to the DECAs of China. This area is to the north of Shanghai, on the southwest side of Changxing Island. The distance between the two sides is ~6–7 km. Ships leave the Yangtze**

River and sail into the East China Sea through this channel. Map data: @MapWorld (http://www.tianditu.gov.cn, last access: 5 March 2020).

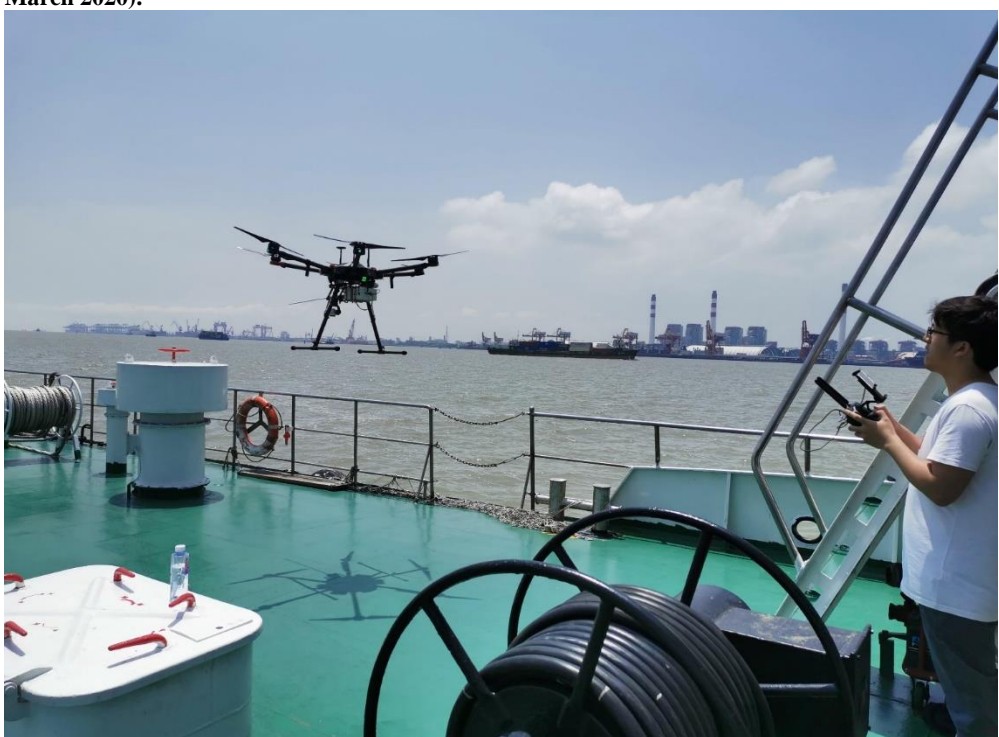

**Figure 3. Operator controlling the takeoff of the UAV from a patrol boat.**

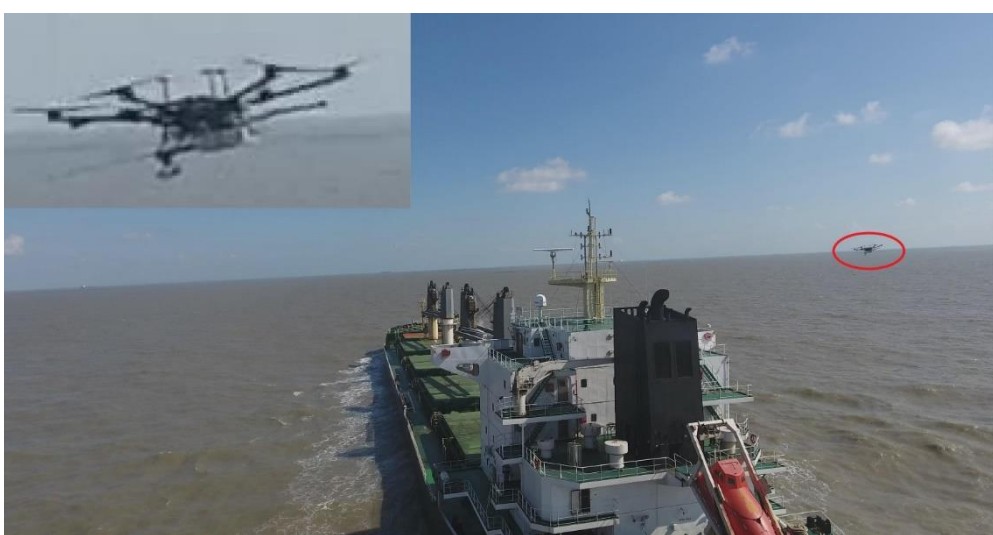

**Figure 4. UAV (marked by a red circle) monitoring a ship's emissions in the open sea. The enlarged UAV is shown in the top left corner. This picture was captured by another UAV.**

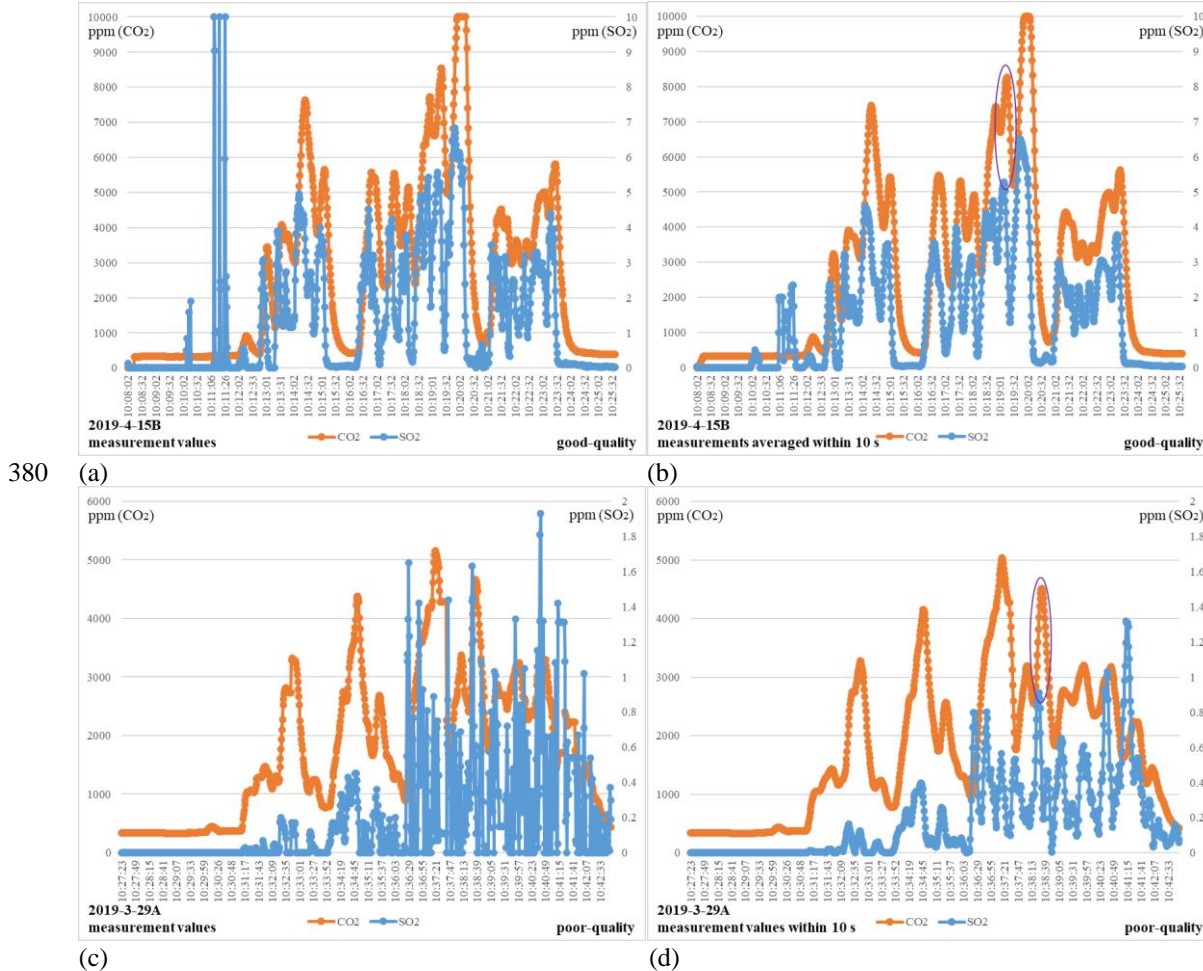

 (a)

(b)

(c)

(d)

**Figure 5. Typical measurement data for SO₂ and CO₂ concentrations, and their corresponding average values within 10 s. (a) and (b) good-quality data from plume ID 2019-4-15B. (c) and (d) poor-quality data from plume ID 2019-3-29A. There are some errors in the measurements from 10:11:06 to 10:12:02 in (a), which may have been caused by sensor uncertainty. These data were ruled out and did not affect the calculation results. After selection, the peak values are circled in purple.**

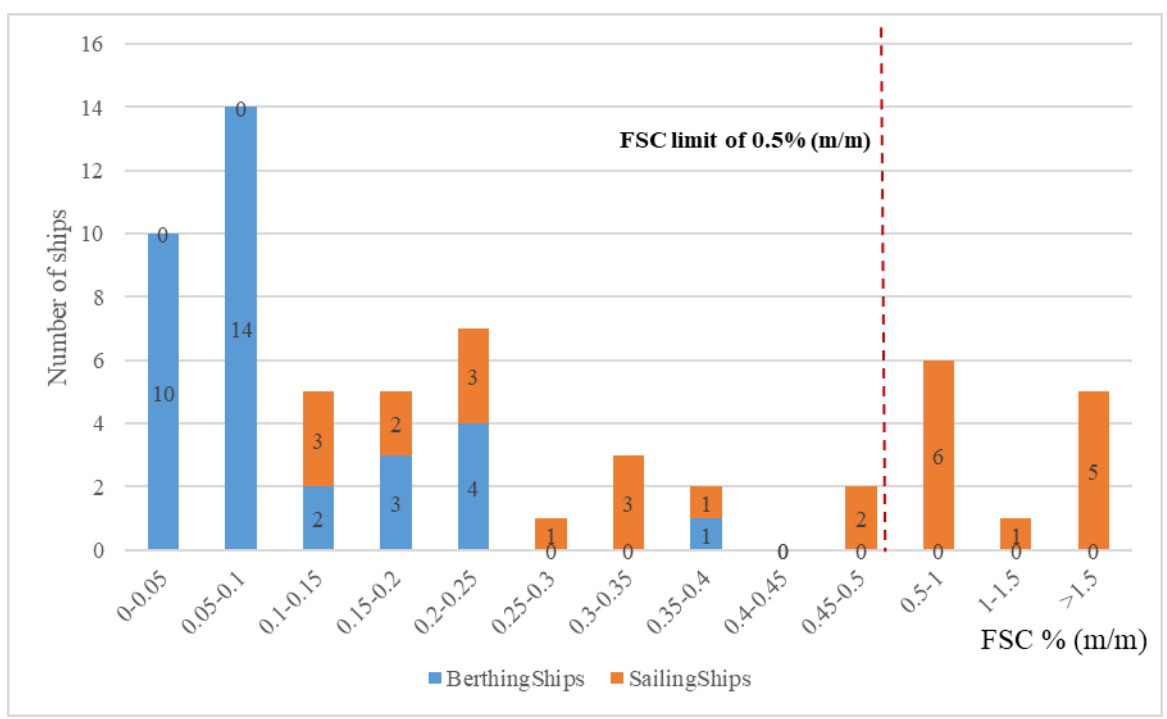

**Figure 6. Comparison between the monitoring results of berthing ships and sailing ships.**

 **Table 1: Parameters of the UAV system.**

|  | Parameter | Value |
|---|---|---|
| **UAV** | Symmetrical motor wheelbase | 1133 mm |
|  | Size | 1668 mm × 1518 mm × 727 mm |
|  | Weight | 9.5 kg |
|  | Recommended maximum take-off weight | 15.5 kg |
|  | Hovering accuracy(P-GPS) | Vertical: ±0.5 m, Horizontal: ±1.5 m |
|  | Maximum rotational angular velocity | Pitch axis: 300°/s, Heading axis: 150°/s |
|  | Maximum pitch Angle | 25° |
|  | Maximum rising speed | 5 m/s |
|  | Maximum rate of descent | 3 m/s |
|  | Maximum sustained wind speed | 8 m/s |
|  | Maximum horizontal flight speed | 65 km/h (no wind environment) |
|  | Hover time | Non-loaded: 32 min; load 6 kg: 16 min |
| **$SO_2$ sensor** | Type | SGA-700A-SO2 |
|  | Principle | Electrochemistry |
|  | Measuring range | 0–10 ppm |
|  | Diameter and height | 33.5 mm; 31 mm |
|  | Weight | 30 g |
|  | Accuracy | $\leq \pm 3$ % (0.3 ppm) |
|  | Linear error | $\leq \pm 2$ % (0.2 ppm) |
|  | Repeatability | $\leq \pm 2$ % (0.2 ppm) |
|  | Power consumption | $\leq 50$ mA |
|  | Response time ($T_{90}$) | $\leq 30$ s |
| **$CO_2$ sensor** | Type | SGA-700A-CO2 |
|  | Principle | Non-Dispersive InfraRed |
|  | Measuring range | 0–10000 ppm |
|  | Diameter and height | 33.5 mm; 31 mm |
|  | Weight | 30 g |
|  | Accuracy | $\leq \pm 3$ % (300 ppm) |
|  | Linear error | $\leq \pm 2$ % (200 ppm) |
|  | Repeatability | $\leq \pm 2$ % (200 ppm) |
|  | Power Consumption | $\leq 100$ mA |
|  | Response time ($T_{90}$) | $\leq 30$ s |

**Table 2: All peak values and their corresponding FSC results. The background values of plume 2019-4-15B were 0 ppm and 310 ppm for SO₂ and CO₂, respectively. The background values of plume 2019-3-29A were 0 ppm and 329 ppm for SO₂ and CO₂, respectively. The remarks indicate the reason for choosing or not choosing the peak. It can be seen that the peak value of plume 2019-4-15B was more obvious and that the results obtained by multiple alternative peaks were similar. The peak of plume 2019-3-29A was less obvious and there were fewer alternative peaks. This was also the basis for distinguishing data as being of a "good"/"poor" quality. The FSC result of selected peak values are marked as "√". As the sensor response time was inconsistent, only the SO₂ peak time points are listed (the CO₂ peak time points had a delay of several seconds).**

| Plume ID | Time point of the SO₂ peak | Peak value of SO₂ and CO₂ (ppm) | Estimated value of FSC (% (m/m)) | True value of FSC (% (m/m)) | Remark |
|---|---|---|---|---|---|
| 2019-4-15B | 10:12:52 | 2.406, 3247 | 0.190 | 0.168 | Reject; less obvious peak values |
| | 10.13.23 | 3.235, 3913 | 0.208 | | |
| | 10.14.07 | 4.594, 7461 | 0.149 | | Non-maximum peaks of alternative peak values |
| | 10.14.57 | 3.529, 5429 | 0.160 | | |
| | 10.16.39 | 3.549, 5475 | 0.159 | | |
| | 10:17:27 | 3.989, 5322 | 0.185 | | |
| | 10:18:01 | 3.159, 4923 | 0.159 | | |
| | 10:18:47 | 4.757, 7430 | 0.155 | | |
| | 10:19:11 | 5.287, 8276 | 0.154 (√) | | Maximum peak of the alternative peak value |
| | 10:19:46 | 6.515, 10000 | 0.156 | | Reject; measurements exceeded the range |
| 2019-3-29A | 10:34:41 | 0.399, 4160 | 0.024 | 0.035 | Reject, less obvious peak values |
| | 10:35:19 | 0.258, 2570 | 0.027 | | Non-maximum peaks of the alternative peak values |
| | 10:37:15 | 0.567, 5036 | 0.028 | | Reject; less obvious peak values |
| | 10:38:27 | 0.913, 4517 | 0.051 (√) | | Maximum peak of the alternative peak value |
| | 10:40:37 | 1.031, 3179 | 0.084 | | Reject; error in the measurement data |
| | 10:41:13 | 1.321, 2254 | 0.159 | | |

**Table 3: Comparison and verification of the estimated (UAV) and true (sampled fuel) values of the FSC from 11 berthing ships.**

| ID | Estimated value of FSC (% (m/m)) | True value of FSC (% (m/m)) | Deviation (% (m/m)) | Quality |
|---|---|---|---|---|
| 2019-3-18A | 0.207 | 0.222 | -0.015 | Good |
| 2019-3-22A | 0.062 | 0.099 | -0.037 | Good |
| 2019-3-22B | 0.046 | 0.042 | 0.004 | Good |
| 2019-3-29A | 0.051 | 0.035 | 0.016 | Poor |
| 2019-4-1A | 0.064 | 0.079 | -0.015 | Good |
| 2019-4-3A | <0.020 | 0.013 | N | Poor |
| 2019-4-3B | 0.052 | 0.092 | -0.040 | Good |
| 2019-4-12A | <0.020 | 0.004 | N | Poor |
| 2019-4-12B | 0.080 | 0.080 | 0 | Good |
| 2019-4-15A | 0.035 | 0.044 | -0.009 | Good |
| 2019-4-15B | 0.154 | 0.168 | -0.014 | Good |

**Table 4: Estimated (UAV) values of the FSC from 27 sailing ships. "*" indicates that the ship was boarded by the maritime authority for inspection, and the value shown in parentheses is the result of the chemical examination of the fuel.**

| ID | Estimated value of FSC (% (m/m)) | Quality | ID | Estimated value of FSC (% (m/m)) | Quality |
|---|---|---|---|---|---|
| 2019-7-12A | 0.634 | Good | 2019-8-22A | 0.178 | Good |
| 2019-7-15A | 0.482 | Good | 2019-8-22B | 0.328 | Poor |
| 2019-7-15B* | 1.563 (0.534) | Good | 2019-8-22C | 0.376 | Good |
| 2019-7-25A | 0.523 | Good | 2019-8-22D | 0.102 | Poor |
| 2019-7-25B | 0.521 | Good | 2019-8-22E | 0.104 | Good |
| 2019-8-14A* | 2.231 (0.744) | Good | 2019-8-22F | 0.232 | Poor |
| 2019-8-15A | 0.305 | Good | 2019-9-17A | 0.196 | Good |
| 2019-8-15B | 0.694 | Poor | 2019-9-17B | 0.567 | Poor |
| 2019-8-16A | 0.137 | Poor | 2019-9-27A | 0.278 | Poor |
| 2019-8-16B | 0.202 | Poor | 2019-9-27B* | 3.449 (1.991) | Good |
| 2019-8-16C | 0.536 | Good | 2019-10-9A | 2.004 | Poor |
| 2019-8-16D | 0.451 | Poor | 2019-10-17A | 0.305 | Good |
| 2019-8-20A | 1.022 | Poor | 2019-10-24A | 0.229 | Good |
| 2019-8-20B* | 3.381 (0.813) | Good | | | |

410