# Peer review of "Monitoring compliance with fuel sulfur content regulations of sailing ships by unmanned aerial vehicle (UAV) measurements of ship emissions in open water"

_Atmospheric Measurement Techniques, 2020_

## Referee Comment (RC1) · J. Duyzer (Referee) · 26 Mar 2020

Journal: AMT Title: Monitoring compliance with fuel sulfur content regulations of sailing ships by unmanned aerial vehicle (UAV) measurements of ship emissions in open water Author(s): Fan Zhou et al. MS No.: amt-2020-12 MS Type: Research article

Referee Report: Jan Duyzer General comments: This is an interesting paper on an important issue. Sulphur compliance monitoring at sea. The method using drones is rather new and the paper shows details that could help other researchers in this

area. Especially the verification measurements presented in the paper and the experiences with the on-board verification attempts are important. Worldwide readers will be interested in the results of the measurements but also in the experiences with on-board inspections. The paper reads well and although it is only a small contribution, I think it deserves publication paying attention to some of the points below. I would like to see a more detailed and perhaps more quantitative treatment of the way the measured data are handled and converted to FSC and when they are rejected. I have only a few small comments on the English (see below). Other more general comments: - The qualifications poor and good are recognizable. We use that as well in our monitoring. Yet I would like to challenge the authors to come up with a more objective assessment of the quality of each measurement or at least some description as to why some measurements are considered poor. It could be difficult to find objective measures, but it seems needed before the method will become a true enforcement tool. - The S-content is now derived from the "fluctuating" signal presented in figure 5. It is not entirely clear why the 10 s averaged data would lead to a better result. I can hardly see the difference. Please show why this is better - It should be noted that the signal is not noisy but simply reflects the incomplete mixing of the exhaust gas with clean air. The peaks represent the air that is exhausted from the funnel i.e. only in the peaks you will find the ratio between SO2 and CO2 that is a direct measure of the fuel composition. The highest peak is probably the best choice but could still be a result of mixing of clean air with exhaust air and lead to bias in the result. Or is this negligible? I would welcome a discussion showing that the peak height is a good measure. - Why not convert table 1 and table 2 into x-y graphs? Perhaps if combined? - Our enforcements contacts tell us that ship owners will normally use fuel with a Sulphur content just below the limit. If I look at all the individual samples, I don't see that. Is there an uncertainty that is missed or are the vessels changing from one fuel to the other at the time of the sampling? More specific comments: Abstract: Line 12 Emissions of CO2 and SO2 are not measured if I am correct. S content (S%) is measured. Line 13: I don't think the costs of this method

are presented or discussed explicitly in this paper. The cost of a vessel capable of operating in open sea seems neglected. In our country that is not low cost. This is also rather costly. Line 17 According to the monitoring results: I suggest changing to: Based upon the online monitoring results …. Line 69: low cost but doesn't include the cost of sailing Line 90 precision of 5 % at full range (is 10 ppm) or 0.5 ppm. Is that correct? Please mention. And how high is that compared to the observed values? What is the Sulphur content of the example presented in figure 5? Please add. Line 90 etc. Could some details or results of the calibration procedure be presented as well. Line 107 "measure the concentration of SO2 and CO2". Change to: Measure the concentration of SO2 and CO2 in the plume Line 124 EF has no unit or? I wonder why equation 1 is mentioned. I think it is a bit confusing. Line 132: What is sampling rate? Electronically? And why are the SO2 and CO2 sensors not synchronized? In the graphs it looks like a delay. And you could just shift them a little. Why is that not done? And why is the 10 s averaged data better. I don't see that. Figure 5 Please use equal y-scales in the right and left panel. This is confusing. Line 137: The 10 sec averages hardly differ from 1 sec data. How could that happen? Line 140 What is meant by the calculated function? Not just the average measurement value? Line 160: What would be objective criteria to tell whether it's a poor-quality plume? Line 167 Is this correct 20% (m/m)? I would expect 20% uncertainty with no units. So I suggest to leave (m/m) out. Line 168 and 169. Unclear what is meant here (after therefore)? It could be interesting for the reader interested in enforcement to mention (even in the abstract) how many of the Non Compliants were detected and how many were missed (i.e. ships that were not identified as Non Compliants but had an FSC above the limit. Line 189: perhaps the word optimistic is not the right word. Perhaps it should be: The uncertainty in the assessment is not small but the results so far, do not lead to optimism with respect to the FSC used by ships sailing in the area. Line 210: It Is only a small sample isn't it, but still convincing looking at figure 6. Please mention that. Page 15: Number of digits in the given numbers are large (such as 40913 ppm, I suggest changing that to 4.1 %) Page 15 Why are the results of the sampling by the

maritime authority not given in the table English: Line 15 ships → vessels Line 45 and 66 supervise or supervision→ enforce and enforcement Line 46 Several studies have suggested monitoring methods or similar. Otherwise I understood wrong but then the sentence is not very clear. Line 54 and 65 navigation → navigating (?) Line 60: airplace→ aircraft Line 65 inaccurate → non representative Line 67 Suggestion: leave therefore out Line 85 extracts gas? → draws air line 108: approximately a few hundred meters. This is double: a few hundred meters is already an expression showing that it is an approximate value. Suggest leaving the word approximately out. Page 13 legends to figure 4: the enlarged UAV is shown in the top left corner (and it is in the right corner: a detail) NB: I have two versions of the text. Also, one with an appendix including two figures: Figure A2 The Chinese text could be difficult to read for non-Chinese readers. Perhaps add some explanation of the Chinese text.

Please also note the supplement to this comment:
https://www.atmos-meas-tech-discuss.net/amt-2020-12/amt-2020-12-RC1-supplement.pdf

---

## Referee Comment (RC2) · Anonymous Referee #2 · 14 May 2020

The paper by Zhou et al. reports on fuel sulfur content (FSC) compliance monitoring of sailing ships with unmanned aerial vehicles (UAV). Measurements were carried out in the Yangtze River Delta close to Shanghai in China, which is selected as an domestic emission control area (DECA) in China having a FSC limit of 0.5% (m/m). Since measurements of the FSC and therefore compliance monitoring from sailing ships are sparse, the topic is of interest not only for the scientific community. The manuscript is in general clearly written and I recommend it for publication in AMT. However, to better demonstrate the quality of the instrumentation and the methods used in this study

more details should be given in the paper.

Instrumentation:

- There are no details on the custom sensors for SO2 and CO2 given. At least a link to a data sheet (in English) or better a table with specifications is needed.

- I am wondering about the short response time of the modules (T90< 1s) which is much better than every electrochemical sensor I know. Looking to figure 5 this response time is very unlikely. SO2 and CO2 measurements of the same plume are out of phase (at least 10 to 15 s) having also completely different gradients.

- Please give more information on the method water vapor was filtered out. What about contamination with particles/soot . . .?

- Have the authors investigated cross-sensitivities to e.g. NO and NO2?

- Measurements were carried out close to the funnel. Therefore, temperatures of the air sucked into the system are highly variable. How this is accounted for?

- Give information on the calibration methods used in this study to ensure long-term data quality. Since sensors used in this study are completely different to those used in Zhou et al. 2019 a simple plot showing the outcome of the sensors when using standard gas mixtures (e.g. 5 and 0 ppm SO2) would be nice.

- The UAV used in this study is the same as in Zhou et al., 2019. What does it mean for the off-shore measurements? What is the operation time under typical weather conditions having e.g. a wind speed of 5 m/s? What is the maximum reasonable distance to a sailing ship? Please add a table with specifications of the whole UAV system.

Methods and uncertainties:

- The authors used peak values of SO2 and CO2 after applying a running mean of 10 s to the measurements to calculate the FSC. Looking to Figure 5 b it is not clear to

me, how this could give reasonable results. As already mentioned above the gradients (and therefore real response times of the sensors) look completely different even for averaged values. At least one example proving and illustrating this method is needed.

- Simply taking into account the given accuracies for the sensors (5 and 3 % on measuring ranges respectively) an error of at least 0.03 % (m/m) can be calculated. Within this calculation no other errors e.g. due to the measurement procedure are included. Therefore, the reported total uncertainty of 0.03 for low FSC levels sounds quite optimistic to me.

- Please give details or an illustration what is meant with good- and poor-quality data.

Minor corrections:

- 2.1. Instrumentation: I guess, dimensions of the pod are given in cm.

- Figure 5: More details needed, which values are used for the calculation of the FSC (see above)

- Tables 1 and 2: Please always give FSC Values in % (m/m) using only the number of meaningful digits. "True value" implies that the analysis of fuel samples have no error which is of course not the case. Please refer to e.g. fuel sample analysis and give the typical error for this method (should be roughly 0.01 % (m/m))

- Table 2: Add values for the fuel sample analysis when available.

---

## Author Comment (AC1) · 19 Jun 2020

Answer to Referee #1

We would like to thank Referee #1 for his/her positive and constructive comments and suggestions. We have studied comments carefully and made corrections, which we hope meet with approval. Comments and responses are listed as follows. In order to facilitate the reference to the questions and proposed changes, we use the following color coding:

Color coding:

**Referee comment**

Our answer

Proposed change in manuscript
* * *
**This is an interesting paper on an important issue. Sulphur compliance monitoring at sea. The method using drones is rather new and the paper shows details that could help other researchers in this area. Especially the verification measurements presented in the paper and the experiences with the on-board verification attempts are important. Worldwide readers will be interested in the results of the measurements but also in the experiences with on-board inspections. The paper reads well and although it is only a small contribution, I think it deserves publication paying attention to some of the points below. I would like to see a more detailed and perhaps more quantitative treatment of the way the measured data are handled and converted to FSC and when they are rejected. I have only a few small comments on the English (see below).**

Thank you for the comments, we are very encouraged.

**Other more general comments:**
**- The qualifications poor and good are recognizable. We use that as well in our monitoring. Yet I would like to challenge the authors to come up with a more objective assessment of the quality of each measurement or at least some description as to why some measurements are considered poor. It could be difficult to find objective measures, but it seems needed before the method will become a true enforcement tool.**

How to objectively and correctly evaluate the quality of plume data is indeed a very important scientific issue, which is also the key to the effective application of the measurement results for maritime law enforcement. I think it would be ideal to design a computational model with input values for plume measurements, output values for FSC results and corresponding confidence levels (or a score that represents the quality of plume data).

This model requires two aspects of work: 1. Adequate an adequate data sets, including gas measurements and the true value of FSC. However, in many cases the true value of the FSC is not available, especially sailing ships. 2. Design and implement the model based on the data sets.

Therefore, there needs a lot of work to implement this model. In fact, that's what we're working on right now. In our current application, however, we assess data quality primarily on the basis of experience. To illustrate how to evaluate data quality, I have supplemented the manuscript with typical good- and poor- data.

[Figure]

(a)                                                    (b)

[Figure]

(c)                                      (d)

**Figure 5. Typical measurement data for $SO_2$ and $CO_2$ concentrations, and their corresponding average values within 10 s. (a) and (b) good-quality data from plume ID 2019-4-15B. (c) and (d) poor-quality data from plume ID 2019-3-29A. There are some errors in the measurements from 10:11:06 to 10:12:02 in (a), which may have been caused by sensor uncertainty. These data were ruled out and did not affect the calculation results.**

**- The S-content is now derived from the "fluctuating" signal presented in figure 5. It is not entirely clear why the 10 s averaged data would lead to a better result. I can hardly see the difference. Please show why this is better.**

I have added description and discussion in the manuscript as bellow.

The average gas concentration within 10 s was chosen for the FSC calculations; however, this does not mean that 9 s or 11 s could not have been selected. To demonstrate this, a comparison calculation was carried out using both 9 s and 11 s, which showed that these led to very little differences in the results. However, it is necessary to ensure that the gradient of the gas measurements is stable within the sampling time (the interval length of the integral). Moreover, the interval length cannot be too short (e.g., 2 s) or too long (e.g., 20 s). If the time is too short, it is difficult to determine whether the measurements are stable and undisturbed over time. Similarly, if the time is too long, it is also difficult to ensure that all of the measurements in the integral interval are stable and undisturbed. In addition, during the flight of the UAV in this study, the time available for measuring the plume was ~5 minutes. As both the ship and the UAV were moving at this time, it was virtually impossible to ensure that the UAV was flying consistently within the plume and obtaining stable measurements. Accordingly, 10 s is also a relatively appropriate value for the measurement process.

**– It should be noted that the signal is not noisy but simply reflects the incomplete mixing of the exhaust gas with clean air. The peaks represent the air that is exhausted from the funnel i.e. only in the peaks you will find the ratio between SO2 and CO2 that is a direct measure of the fuel composition. The highest peak is probably the best choice but could still be a result of mixing of clean air with exhaust air and lead to bias in the result. Or is this negligible? I would welcome a discussion showing that the peak height is a good measure.**

According to my understanding, the key problem is how to ensure that the gas being measured is evenly mixed. Or, the selected peaks of $SO_2$ and $CO_2$ are measured from the complete mixing of the exhaust gas with clean air. And then the ratio between $SO_2$ and $CO_2$ can be used to calculate the fuel composition. Assuming that the gas is mixed complete, the variation trend of $SO_2$ and $CO_2$ measurements should be the same (given that the corresponding time of the sensor is not consistent, there may be some deviation), and this trend can be easily identified in the peak area.

But not all peaks can be used to calculate the FSC. Therefore, in the previous study, we developed a selection process. The measurements from incomplete mixing have been ruled out. Meanwhile, the measurements with errors have also been ruled out. The maximum values are likely to have been measured in the center of the ship's plume. At that location, the measurement value is relatively stable, and the probability of interference from other factors is lower. At the same time, the higher the peak value, the greater the proportion of exhaust gas, so the impact from the incomplete mixing of

the exhaust gas with clean air is smaller.

To sum up, the obvious and stable maximum peak appears in the measured value over 10 s periods as the calculated value is a more appropriate choice. There are, of course, cases where multiple similar peaks can occur simultaneously. At this time, their calculations may be very similar, in which case, the results obtained by the calculation of the highest peak should have high credibility. However, it is difficult to describe the problem quantitatively and further research is needed. The ideal model that I mentioned above need a lot of work to do. I have added description and discussion in the manuscript as bellow.

Nevertheless, there is also some uncertainty associated with choosing the peak values. After ruling out the peak values across the full range as well as those corresponding to dramatic changes, the global maximum values were selected as the peak values to calculate the FSC. The maximum values probably correspond to the measurements taken in the center of the ship's plume. At that location, the measurement values were relatively stable, and the probability of interference from other factors was lower. Furthermore, the higher the peak value is, the greater the proportion of exhaust gas is; hence, the impact from the incomplete mixing of the exhaust gas with clean air is relatively small.

In summary, the obvious and stable maximum values are selected as peak values to calculate the FSC. There are, of course, situations where multiple similar peaks can occur simultaneously. In this case, their calculated FSCs may be very similar, and the results obtained by the calculation of the highest peak should have high credibility, for instance, the measurements of plume 2019-4-15B.

I also added the selection of peak values of Figure 5 in the Table 2 as bellow.

**Table 2: All peak values and their corresponding FSC results. The background values of plume 2019-4-15B were 0 ppm and 310 ppm for $SO_2$ and $CO_2$, respectively. The background values of plume 2019-3-29A were 0 ppm and 329 ppm for $SO_2$ and $CO_2$, respectively. The remarks indicate the reason for choosing or not choosing the peak. It can be seen that the peak value of plume 2019-4-15B was more obvious and that the results obtained by multiple alternative peaks were similar. The peak of plume 2019-3-29A was less obvious and there were fewer alternative peaks. This was also the basis for distinguishing data as being of a "good"/"poor" quality. The FSC result of selected peak values are marked as "√".**

| Plume ID | Time point | Peak value of $SO_2$ and $CO_2$ (ppm) | Estimated value of FSC (% (m/m)) | True value of FSC (% (m/m)) | Remark |
|---|---|---|---|---|---|
| 2019-4-15B | 10:12:52 | 2.406, 2020 | 0.326 | 0.168 | Reject; less obvious peak values |
| | 10.13.23 | 3.235, 2372 | 0.364 | | |
| | 10.14.07 | 4.594, 4665 | 0.245 | | Non-maximum peaks of alternative peak values |
| | 10.14.57 | 3.529, 4872 | 0.179 | | |
| | 10.16.39 | 3.549, 4444 | 0.199 | | |
| | 10:17:27 | 3.989, 3911 | 0.257 | | |
| | 10:18:01 | 3.159, 4607 | 0.171 | | |
| | 10:18:47 | 4.757, 6895 | 0.168 | | |
| | 10:19:11 | 5.287, 7634 | 0.167 (√) | | Maximum peak of the alternative peak value |
| | 10:19:46 | 6.515, 8100 | 0.194 | | Reject; measurements exceeded the range |
| 2019-3-29A | 10:34:41 | 0.399, 3880 | 0.026 | 0.035 | Reject, less obvious peak values |
| | 10:35:19 | 0.258, 2011 | 0.036 | | Non-maximum peaks of the alternative peak values |
| | 10:37:15 | 0.567, 4994 | 0.028 | | Reject; less obvious peak values |
| | 10:38:27 | 0.913, 4022 | 0.057 (√) | | Maximum peak of the alternative peak value |
| | 10:40:37 | 1.031, 2996 | 0.090 | | Reject; error in the measurement data |
| | 10:41:13 | 1.321, 1700 | 0.224 | | |

**According to my understanding, the above comments are all about the 2.4 Calculation and 2.5 Uncertainties. I have rewritten these parts.**

**- Why not convert table 1 and table 2 into x-y graphs? Perhaps if combined?**

Please note that the estimated (UAV) and true (sampled fuel) values of the FSC from 11 berthing ships are list in table 1. But there are only estimated (UAV) values of the FSC from 27 sailing ships are list in table 2. It seems that table 2 cannot convert into x-y graphs and the table 1 and table 2 cannot be combined.

**- Our enforcements contacts tell us that ship owners will normally use fuel with a Sulphur content just below the limit. If I look at all the individual samples, I don't see that. Is there an uncertainty that is missed or are the vessels changing from one fuel to the other at the time of the sampling?**

I think it's different in different areas, especially when the regulations are different.
In the DECA of China, the regulations are as follow:
Starting January 1, 2017, the FSC cannot exceed 0.5% (m/m) during berthing, excluding the first hour after arrival and the last hour before departure.
Starting January 1, 2018, the FSC cannot exceed 0.5% (m/m) during berthing.
Starting January 1, 2019, the FSC cannot exceed 0.5% (m/m) for both sailing and berthing ships.
Overall, our FSC monitoring results are shown in Figure 6. It shows that the FSCs of the sailing ships were considerably higher than those of the berthing ships. The FSCs of berthing ships are measured in the year of 2018 and 2019. The FSCs of sailing ships are measured in the year of 2019. On July 15, 2019, it was the first time that a sailing ship had been caught for having failed the FSC regulations in China.
I believe that with the further implementation of the policy, the ship owners will normally use fuel with a Sulphur content just below the limit in the DECA of China.

**More specific comments: Abstract: Line 12 Emissions of CO2 and SO2 are not measured if I am correct. S content (S%) is measured.**

Yes, this sentence "measure the sulfur dioxide and carbon dioxide emissions from sailing ships" is not appropriate. I guess the S content (S%) is calculated from the concentrations of $SO_2$ and $CO_2$. This sentence has been modified as follow.

The present study adopts a monitoring method involving an unmanned aerial vehicle (UAV) that takes off from a patrol boat to measure the concentrations of $SO_2$ and $CO_2$ within the plumes of sailing ships.

**Line 13: I don't think the costs of this method are presented or discussed explicitly in this paper. The cost of a vessel capable of operating in open sea seems neglected. In our country that is not low cost. This is also rather costly.**

Yes, this is rather costly. The patrol boats of the maritime department should cruise in the area regularly (once a week). To keep costs down, we measure the ships plume at the same time. Therefore, the cost is lower compared to aircraft and applicable to maritime department. I have added the discussion in the manuscript.

The method proposed in this study can be used to monitor ship emissions at a comparatively low cost to understand the FSCs of sailing ships in open waters. Although the cost of using patrol boats is not negligible, it is convenient and lower cost for maritime authorities compared with small aircraft.

**Line 17 According to the monitoring results: I suggest changing to: Based upon the online monitoring results ....**
OK, this sentence has been modified.

**Line 69: low cost but doesn't include the cost of sailing**

As mentioned above, I have added discussion in the manuscript.

**Line 90 precision of 5 % at full range (is 10 ppm) or 0.5 ppm. Is that correct? Please mention. And how high is that compared to the observed values? What is the Sulphur content of the example presented in figure 5? Please add.**

Yes, it's relative to the range (0.5 ppm is 5% of 10 ppm full range). According to the comments of Referee #2, I have supplemented the detailed parameter information of the UAS. In the original manuscript, the range of $SO_2$ sensor and accuracy of $CO_2$ sensor were wrong. I have modified and cross-checked to make sure the product information was right.

There are different range models of sensors, such as 5 ppm, 10 ppm, 100 ppm and so on. Precision generally depends on the size of the range. For example, precision of 0.25 ppm for 5 ppm, 0.5 ppm for 10 ppm, and 5 ppm for 100 ppm. It needs to make sure that observed values do not exceed the range in most cases, and the precision should not be too low. Therefore, observed values of $SO_2$ are generally in the range of 0-10 ppm. $CO_2$ are 400-5000 ppm range.
As mentioned above, I have added Table 2.

**Line 90 etc. Could some details or results of the calibration procedure be presented as well.**

I have added the details in the manuscript.

These sensor characteristics were provided by the instrument manufacturer and were ensured to be within the tolerances by calibration. The zero and full scales are usually calibrated by a standard mixed gas when the equipment is used on a daily basis. The major parameters of the UAS are listed in Table 1.

**Line 107 "measure the concentration of SO2 and CO2". Change to: Measure the concentration of SO2 and CO2 in the plume**

OK, this sentence has been modified.

**Line 124 EF has no unit or? I wonder why equation 1 is mentioned. I think it is a bit confusing.**

EF has unit, $g_{SO2}/kg_{fuel}$, in g emitted per kg fuel. I have added it in manuscript.
Equation 1 is a description of the measurement principle. We use the parameter of 10s to derive equation 2. If don't discuss equation 1, just list equation 2. It seems too sudden to the reader.

**Line 132: What is sampling rate? Electronically? And why are the SO2 and CO2 sensors not synchronized? In the graphs it looks like a delay. And you could just shift them a little. Why is that not done? And why is the 10 s averaged data better. I don't see that. Figure 5 Please use equal y-scales in the right and left panel. This is confusing.**

The sampling rate is 1s.
This is due to inconsistent response times of different sensors. In the vast majority of cases, the response time of $SO_2$ are faster, and the $CO_2$ looks a little bit behind $SO_2$.
Of course, we can adjust the timing artificially. For example, adjust the peak time of $CO_2$ to the peak time of $SO_2$.
I've done a comparative calculation. The data set is of the estimated (UAV) and true (sampled fuel) values of the FSC

from 11 berthing ships (using second-generation pod). On the whole, the difference is not great and the accuracy was slightly reduced. Therefore, we chose the method described in the manuscript to do the calculation.

The average gas concentration over 10 s was chosen for the FSC calculations; however, this does not mean that 9 s or 11 s could not have been selected. To demonstrate this, a comparison calculation was carried out using both 9 s and 11 s, which showed that these led to very little differences in the results. However, it is necessary to ensure that the gradient of the gas measurements is stable over the sampling time (the interval length of the integral). Moreover, the interval length cannot be too short (e.g., 2 s) or too long (e.g., 20 s). If the time is too short, it is difficult to determine whether the measurements are stable and undisturbed over time. Similarly, if the time is too long, it is also difficult to ensure that all of the measurements in the integral interval are stable and undisturbed. In addition, during the flight of the UAV in this study, the time available for measuring the plume was ~5 minutes. As both the ship and the UAV were moving at this time, it was virtually impossible to ensure that the UAV was flying consistently within the plume and obtaining stable measurements. Accordingly, 10 s is also a relatively appropriate value for the measurement process.

Therefore, 10 s is an appropriate parameter.

I have added the details in the manuscript as mentioned above.

OK, I have added equal y-scales in the right and left panel.

**Line 137: The 10 sec averages hardly differ from 1 sec data. How could that happen?**

Does this refer to the data in Figure 5?

The data in Figure 5 (b) is relatively smooth, making it easy to select peak values.

The data in Figure 5 is of good-quality and the difference is not significant. This is even more pronounced if the data is of poor-quality. I've added good- and poor- typical figures as mentioned above.

**Line 140 What is meant by the calculated function? Not just the average measurement value?**

It is "calculate the average of the data within 10 s". This sentence has been modified.

**Line 160: What would be objective criteria to tell whether it's a poor-quality plume?**

In the previous study, we developed a selection process. If most of the data in the measurement dataset is ruled out, then it is a poor-quality plume. If multiple peaks are preserved and their FSC result are similar, then it is a good-quality plume. But, strictly speaking, we do not have a completely objective evaluation criterion. This requires the computational model I mentioned above. I have added tow typical example in the Figure 5 and Table 2.

**Line 167 Is this correct 20% (m/m)? I would expect 20% uncertainty with no units. So I suggest to leave (m/m) out.**

Yes, this sentence has been modified.

**Line 168 and 169. Unclear what is meant here (after therefore)? It could be interesting for the reader interested in enforcement to mention (even in the abstract) how many of the Non Compliants were detected and how many were missed (i.e. ships that were not identified as Non Compliants but had an FSC above the limit.**

Please note that boarding inspection is a complicated process. The target ship cannot stop immediately in the channel for inspection and have to sail to the anchorage. Patrol boats need to follow to the anchorage for boarding inspection. In the process, to avoid punishment, crew will take various measures to drain the high-sulfur fuel in the main engine fuel oil

pipeline.

The whole process of boarding inspection requires more than half a day and the work of more than a dozen law enforcement officers (such as sailors, driver, inspectors, VTS watchmen).

Therefore, in order to ensure that can accurately tracked down the offending ship. Law enforcement officers of the Pudong maritime safety administration only intercepted four sailing ships for which the UAV FSC results were of a good-quality and all exceeded 2% (m/m). The chemical FSC results of the four ships were: 0.534% (m/m), 0.744% (m/m), 0.813% (m/m), and 1.991% (m/m).

If the UAV FSC results is just over 0.5%, and then we board the ship for inspection. I believe the Non Compliants and Missed will happen. However, the factor of switching fuel cannot be ignored. Intercept and boarding a sailing ship are not just an experiment.

**Line 189: perhaps the word optimistic is not the right word. Perhaps it should be: The uncertainty in the assessment is not small but the results so far, do not lead to optimism with respect to the FSC used by ships sailing in the area.**

OK, this sentence has been modified.

**Line 210: It Is only a small sample isn't it, but still convincing looking at figure 6. Please mention that.**

OK, I have added the this in the manuscript.

**Page 15: Number of digits in the given numbers are large (such as 40913 ppm, I suggest changing that to 4.1 %)**

OK, these words have been modified.

**Page 15 Why are the results of the sampling by the maritime authority not given in the table.**

OK, I have added this.

**English:**
**Line 15 ships → vessels**
The word "ships" or "vessels" seems all appropriate. For consistency, "ships" is used all over the text.
**Line 46 Several studies have suggested monitoring methods or similar. Otherwise I understood wrong but then the sentence is not very clear.**
I have changed the sentence "Several studies have suggested monitoring ship emissions to estimate the FSC of the target ship" as "Several studies have suggested estimating FSC by measuring ship plumes".
**Line 45 and 66 supervise or supervision→ enforce and enforcement; Line 54 and 65 navigation → navigating (?); Line 60: airplace→ aircraft; Line 65 inaccurate → non representative; Line 67 Suggestion: leave therefore out Line 85 extracts gas? → draws air; Line 108: approximately a few hundred meters. This is double: a few hundred meters is already an expression showing that it is an approximate value. Suggest leaving the word approximately out.; Page 13 legends to figure 4: the enlarged UAV is shown in the top left corner (and it is in the right corner: a detail)**
**The above English grammar problems have been revised. Thank you very much for your earnest help in pointing out the English problems.**
**NB: I have two versions of the text. Also, one with an appendix including two figures: Figure A2 The Chinese text could be difficult to read for non-Chinese readers. Perhaps add some explanation of the Chinese text.**
As the suggestion of the Associate Editor, "remove the figures from the appendix as they do not contribute to the

discussion of the measurement techniques". I have removed the Figure A1 and A2.

---

## Author Comment (AC2) · 19 Jun 2020

Answer to Referee #2

We would like to thank Referee #2 for his/her positive and constructive comments and suggestions. We have studied comments carefully and made corrections, which we hope meet with approval. Comments and responses are listed as follows. In order to facilitate the reference to the questions and proposed changes, we use the following color coding:

Color coding:

**Referee comment**

Our answer

Proposed change in manuscript

**The paper by Zhou et al. reports on fuel sulfur content (FSC) compliance monitoring of sailing ships with unmanned aerial vehicles (UAV). Measurements were carried out in the Yangtze River Delta close to Shanghai in China, which is selected as an domestic emission control area (DECA) in China having a FSC limit of 0.5% (m/m). Since measurements of the FSC and therefore compliance monitoring from sailing ships are sparse, the topic is of interest not only for the scientific community. The manuscript is in general clearly written and I recommend it for publication in AMT. However, to better demonstrate the quality of the instrumentation and the methods used in this study more details should be given in the paper.**

Thank you for the comments, we are very encouraged.

**Instrumentation:**
**- There are no details on the custom sensors for SO2 and CO2 given. At least a link to a data sheet (in English) or better a table with specifications is needed.**

OK, I have added a detail table (Table 1).

**- I am wondering about the short response time of the modules (T90< 1s) which is much better than every electrochemical sensor I know. Looking to figure 5 this response time is very unlikely. SO2 and CO2 measurements of the same plume are out of phase (at least 10 to 15 s) having also completely different gradients.**

We don't make sensors, and sensors were purchased from Shenzhen Singoan Electronic Technology Co., Ltd., China.
I looked up the relevant materials (Mellqvist et al., 2017), the $t_{90}$ of $CO_2$ sensor based on NDIR (LI-COR 7200) is 0.1s. I also consulted other relevant literature (Alföldy et al., 2013, Beecken et al., 2014, Balzani Lööv et al., 2015), the $t_{90}$ of $CO_2$ sensor is about < 1-5 s. The $CO_2$ sensor used by us is also base on NDIR principle. It seems reasonable that the $CO_2$ response time is less than 1 s. However, it can be clearly seen from Figure 5 that the response time of $SO_2$ is significantly faster than that of $CO_2$. Does this mean that the $t_{90}$ of $SO_2$ sensor is also <1s?
To ensure the accuracy of parameter information, I contacted one technical support of Shenzhen Singoan Electronic Technology Co., Ltd., China. I double confirm the relevant parameter information of the sensor. I got the detailed information sheet of these two sensors. The response time (defined as $T_{90}$) is <30 s. "The $t_{90}$ represents the time taken to reach 90% of the stable response following a step change in the sample concentration". I guess "a step change" does not mean from 0 to full range. If it is from 0 to full range, the response time ($T_{90}$) is <30 s. The information is list as follow.

[Figure]

Figure picture of the sensor

The type of $SO_2$ sensor is SGA-700A-$SO_2$. Link: http://www.singoan.com/article/detail/id/6233.htm
The type of $CO_2$ sensor is SGA-700A-$CO_2$. Link: http://www.singoan.com/SGA_400_700_CO2.htm

I fill the main information into Table 1 as bellow. In the original manuscript, the range of $SO_2$ sensor and accuracy of $CO_2$ sensor are wrong. I have modified and cross-checked to make sure the information was consistent as the product information.

**Table 1: Parameters of the UAS**

| | Parameter | Value |
|---|---|---|
| **UAV** | Symmetrical motor wheelbase | 1133 mm |
| | Size | 1668 mm × 1518 mm × 727 mm |
| | Weight | 9.5 kg |
| | Recommended maximum take-off weight | 15.5 kg |
| | Hovering accuracy(P-GPS) | Vertical: ±0.5 m, Horizontal: ±1.5 m |
| | Maximum rotational angular velocity | Pitch axis: 300°/s, Heading axis: 150°/s |
| | Maximum pitch Angle | 25° |
| | Maximum rising speed | 5 m/s |
| | Maximum rate of descent | 3 m/s |
| | Maximum sustained wind speed | 8 m/s |
| | Maximum horizontal flight speed | 65 km/h (no wind environment) |
| | Hover time | Non-loaded: 32 min; load 6 kg: 16 min |
| **$SO_2$ sensor** | Type | SGA-700A-SO2 |
| | Principle | Electrochemistry |
| | Measuring range | 0–10 ppm |
| | Diameter and height | 33.5 mm; 31 mm |
| | Weight | 30 g |
| | Accuracy | ≤ ±3 % (0.3 ppm) |
| | Linear error | ≤ ±2 % (0.2 ppm) |
| | Repeatability | ≤ ±2 % (0.2 ppm) |
| | Power consumption | ≤ 50 mA |
| | Response time ($T_{90}$) | ≤ 30 s |
| **$CO_2$ sensor** | Type | SGA-700A-CO2 |
| | Principle | Non-Dispersive InfraRed |
| | Measuring range | 0–10000 ppm |
| | Diameter and height | 33.5 mm; 31 mm |
| | Weight | 30 g |
| | Accuracy | ≤ ±3 % (300 ppm) |
| | Linear error | ≤ ±2 % (200 ppm) |
| | Repeatability | ≤ ±2 % (200 ppm) |
| | Power Consumption | ≤ 100 mA |
| | Response time ($T_{90}$) | ≤ 30 s |

Alföldy, B., Lööv, J. B., Lagler, F., Mellqvist, J., Berg, N., Beecken, J., Weststrate, H., Duyzer, J., Bencs, L., Horemans, B., Cavalli, F., Putaud, J.-P., Janssens-Maenhout, G., Csordás, A. P., Van Grieken, R., Borowiak, A., and Hjorth, J.: Measurements of air pollution emission factors for marine transportation in SECA, Atmos. Meas. Tech., 6, 1777–1791, https://doi.org/10.5194/amt-6-1777-2013, 2013.

Balzani Lööv, J. M., Alfoldy, B., Gast, L. F. L., Hjorth, J., Lagler, F., Mellqvist, J., Beecken, J., Berg, N., Duyzer, J., Westrate, H., Swart, D. P. J., Berkhout, A. J. C., Jalkanen, J.-P., Prata, A. J., van der Hoff, G. R., and Borowiak, A.: Field test of available methods to measure remotely SOx and NOx emissions from ships, Atmos. Meas. Tech., 7, 2597–2613, https://doi.org/10.5194/amt-7-2597-2014, 2014.

Mellqvist, J., Conde, V., Beecken, J., and Ekholm, J.: Certification of an aircraft and airborne surveillance of fuel sulfur content in ships at the SECA border, CompMon (https://compmon.eu/, last access: 6 November 2018), 2017b.

Beecken, J., Mellqvist, J., Salo, K., Ekholm, J., and Jalknen, J.-P.: Airborne emission measurements of SO2, NOx and particles from individual ships using a sniffer technique, Atmos. Meas. Tech., 7, 1957–1968, http://dx.doi.org/10.5194/amt-7-1957-2014, 2014.

**- Please give more information on the method water vapor was filtered out. What about contamination with particles/soot ...?**

The pod is a convenient, lightweight device. It does not have the same complex gas filtration as shore-based equipment. A hose filter valve was used to filter out the water vapor, particles and soot. Its figure is as follow.

[Figure]

Figure Different type of the hose filter valve, the length is about 4-20mm

**- Have the authors investigated cross-sensitivities to e.g. NO and NO2?**

For the sake of lightweight and convenience, the second-generation pod is only equipped with $SO_2$ and $CO_2$ sensors. In the research of Mellqvist et al. (2017), they proposed a treatment for cross-sensitivities $SO_2$ and $NO_2$. But their sensor is based on the principle of fluorescence (Thermo 43i-TLE), and our sensor is based on electrochemistry. I have looked up the relevant materials, the cross-sensitivities of $SO_2$ and $NO_2$ do exist when using the electrochemical sensor. I guess more experiments and researches are needed to eliminate the effects of cross-induction.

Mellqvist, J., Conde, V., Beecken, J., and Ekholm, J.: Certification of an aircraft and airborne surveillance of fuel sulfur content in ships at the SECA border, CompMon (https://compmon.eu/, last access: 6 November 2018), 2017.

**- Measurements were carried out close to the funnel. Therefore, temperatures of the air sucked into the system are highly variable. How this is accounted for?**

The UAS has a gas pump, gas circuit, filter which can cool the gas to a certain extent.
I know that the equipment used in relevant research work has a constant temperature and pressure detecting environment. The weight of the equipment is usually tens of kilograms. But there are limits to the weight and size of the equipment that UAV can carry.
The conversion relation of units is as follow:

$$\mathbf{X}\ (mg/m^3) = \frac{\mathbf{M}}{22.4} * \mathbf{Y}\ (ppm) * \left[\frac{273}{273 + \mathbf{T}}\right] * (\mathbf{Ba}/101325)$$

M is molecular weight of the gas, T is the temperature, and Ba is air pressure.
I guess it has to redesign the pod and laboratory experiments to reduce the impact of this factor.

**- Give information on the calibration methods used in this study to ensure long-term data quality. Since sensors used in this study are completely different to those used in Zhou et al. 2019 a simple plot showing the outcome of the sensors when using standard gas mixtures (e.g. 5 and 0 ppm SO2) would be nice.**

The calibration process is the same, I have added description.

The details of calibration are as follows: We bought the sensors and then designed and built the pod. Then we send the pod to a third-party inspection agency for certification. Content of verification is mainly about the accuracy. We only know the results and we don't have detailed data. I know that you may interest in the sensors. But the information available to me is limited. I have attached the validation report at the end (the original is in Chinese).

**- The UAV used in this study is the same as in Zhou et al., 2019. What does it mean for the off-shore measurements? What is the operation time under typical weather conditions having e.g. a wind speed of 5 m/s? What is the maximum reasonable distance to a sailing ship? Please add a table with specifications of the whole UAV system.**

The UAV is a product of DJI and is a relatively common UAV model. We have successfully applied it offshore measurements. But there are still have limitations, it is more affected by the weather compare with other measurement platform. For safety reasons, we can't use it when it rains or when the wind is high. Nevertheless, this is basically the best and most suitable civilian UAV that we can find on the market.
About 15-20 min.
This question is hard to say. We did not measure the distance by instruments. At sea, the human sense of distance is very weak. My personal feeling is about 500 to 3000 meters.
I have added the Table 1.

**Methods and uncertainties:**
**- The authors used peak values of SO2 and CO2 after applying a running mean of 10s to the measurements to calculate the FSC. Looking to Figure 5 b it is not clear to me, how this could give reasonable results. As already mentioned above the gradients (and therefore real response times of the sensors) look completely different even for averaged values. At least one example proving and illustrating this method is needed.**

The data in Figure 5 is of good-quality and the difference is not significant. This is even more pronounced if the data is of poor-quality. I've added good and poor typical figures and related discussion. To illustrate how to evaluate data quality, I have supplemented the manuscript with typical good- and poor- data, and the relevant description as follow.

[revised manuscript text omitted]

**I have rewritten these parts of 2.4 Calculation and 2.5 Uncertainties to illustrate the problem and related problem.**

**- Simply taking into account the given accuracies for the sensors (5 and 3 % on measuring ranges respectively) an error of at least 0.03 % (m/m) can be calculated. Within this calculation no other errors e.g. due to the measurement procedure are included. Therefore, the reported total uncertainty of 0.03 for low FSC levels sounds quite optimistic to me.**

According to the literature available, the main method to measure the ship plume are land-based and airborne-based method. UAV measurements are indeed more accurate than these approaches. I guess this is mainly because UAV measurements are taken at close range. However, please note that this accuracy is the measurement result of the berthing ships. One is unable to obtain samples of fuel from sailing ships normally. We attempted to measure more than 40 ship plumes in open water; however, only 27 of them resulted in good- or poor-quality data, i.e., usable data. The success rate is not very high.

**- Please give details or an illustration what is meant with good- and poor-quality data.**

As mentioned above, I have added good- and poor-quality.

**Minor corrections:**
**- 2.1. Instrumentation: I guess, dimensions of the pod are given in cm.**

Yes, you are right, it is cm. Thank you for pointing out this error.

**- Figure 5: More details needed, which values are used for the calculation of the FSC (see above)**

As mentioned above, I have added.

- Tables 1 and 2: Please always give FSC Values in % (m/m) using only the number of meaningful digits. "True value" implies that the analysis of fuel samples have no error which is of course not the case. Please refer to e.g. fuel sample analysis and give the typical error for this method (should be roughly 0.01 % (m/m))

Yes, I have modified the unit.
Maritime authorities send fuel samples to third-party testing institutions, and the test results can be used as a basis for law enforcement. I don't have the information about the error of their detection methods. But I guess the accuracy of direct fuel detection is definitely far higher than the FSC result estimated from gas measurement. It is appropriate to take it as "True value".

**- Table 2: Add values for the fuel sample analysis when available.**

OK, I have added.

**The following is the third-party inspection report of the pod.**

**上 海 市 计 量 测 试 技 术 研 究 院**
**华 东 国 家 计 量 测 试 中 心**
**中 国 上 海 测 试 中 心**

**检 测 报 告**

**Test Report**

| | |
|---|---|
| 委 托 者
Customer | 上海安馨信息科技有限公司 |
| 委托者地址
Address of customer | 上海临港新城海基六路218弄13号楼3楼 |
| 样 品 名 称
Name of sample | 船舶尾气检测吊舱 |
| 制 造 厂
Manufacturer | 上海安馨信息科技有限公司 |
| 型 号 / 规 格
Model/Specification | AX-YD |
| 样 品 编 号
No. of sample | 1145003 |

| | |
|---|---|
| 批 准 人 / 职 务
Approved by / Functions | 郝玉红 质量主管 |
| （机构检测专用章）  核 验 员
Checked by | 核验员标记 |
| 检 测 员
Tested by | 检定员标记 |

检测日期 2019 年 03 月 11 日
Date for test  Year  Month  Day

地址：上海市张衡路1500号(总部) 电话：021-38839800 传真：021-50798390 邮编：201203
Address No.1500 Zhangheng Road,Shanghai(headquarters) Tel. Fax PostCode
客户咨询电话：800-820-5172 投诉电话：021-50798262
Inquire line Tel. for complaint

未经本院/中心批准，部分采用本证书内容无效。
Partly using this report will not be admitted unless allowed by SIMT.

国家法定计量检定机构计量授权证书号(中心/院):(国)法计(2017)01039号/(2017)01019号
The number of the Certificate of Metrological Authorization to The Legal Metrological Verification Institution is No.(2017）01039/ No.（2017）01019
* * *
本次检测所依据的技术规范（代号、名称）：
Reference documents for the test (code 、name)

参照 JJG 635-2011《一氧化碳、二氧化碳红外气体分析器检定规程》
参照 JJG 551-2003《二氧化硫气体检测仪检定规程》
* * *
本次检测所使用的主要测量仪器：
Main measuring instruments used in this test

| 名称
Name | 型号规格
Model | 编号
Number | 测量范围
Measurement range | 不确定度或准确度等级或最大允许误差
Uncertainty/Accuracy Class/Maximum Permissible Error | 证书编号/
有效期限
Certificate No./Due date |
|---|---|---|---|---|---|
| 氮中二氧化碳气体标准物质 | / | 770357/201803 | $4.97×10^{-2}$mol/mol | $U_{rel}=2.0\%$($k=2$） | 770357/
2019-03-29 |
| 氮中二氧化硫标准气 | / | L00502040 | $100×10^{-6}$mol/mol | $U_{rel}=2.0\%$($k=2$) | L00502040/
2019-03-29 |
| 精密气体稀释仪 | MGB1000 | 06132 | 稀释比：1~1000 | ±3% | 2019I30-30-1727259001/
2020-02-14 |
| / | / | / | / | / | / |

检测地点及环境条件：
Location and environmental condition for the test

地点： 张衡路1500号理化东楼125室
Location

温度： 21℃        湿度： 63%RH        其它： /
Ambient temperature        Relative humidity        Others
* * *
备注： /
Note:
* * *
本报告提供的结果仅对本次被测的样品有效。
The data are valid only for the sample(s).
* * *
检测结果/说明：
Results of test and additional explanation

| 委托日期 | 2019.03.07 | 样品状态描述 | 正常 | 受样方式 | 客户送样 |
|---|---|---|---|---|---|

| 检测项目 | 标准气体浓度（μmol/mol） | 仪器显示值（μmol/mol） |
|---|---|---|
| $CO_2$ | 10.0 | 9.84 |
| | 900 | 924 |
| $SO_2$ | 1.0 | 0.96 |
| | 8.0 | 8.21 |

检测结果内容结束

---

## Author Response (AR2)

Dear Andreas Richter,

5

We have studied comments carefully and made corrections, which we hope meet with approval. Comments and responses are listed as follows. In order to facilitate the reference to the questions and proposed changes, we use the following color coding:

**Color coding: Referee comment Our answer Proposed change in manuscript**

As pointed out by one of the reviewers, there is a clear difference in the time constant of the SO2 and CO2 sensors.

- 10 This is less visible in the revised manuscript as you changed the scale but this is something that you need to address. From what I see in your original graphs, the SO2 instrument has a much shorter time constant and thus shows more variability and a time shift compared the CO2 instrument. This is of course a problem when computing FSC if the measurement time in the centre of the plume is short. In such conditions, the SO2 peak will be larger than the corresponding peak in CO2 and the FSC will be overestimated. I think the proper approach would be to
- 15 either take averages long enough to make sure that the full signal is taken by both instruments (30 seconds) or to numerically degrade the SO2 time series to correspond to the time constant of the CO2 instrument.

"take averages long enough to make sure that the full signal is taken by both instruments (30 seconds)". Indeed, this is a very important question. If the time is too long (30 s), it is difficult to ensure that all of the measurements in the integral interval are stable and undisturbed, especially for poor-quality data. In addition, the length of the valid data (especially  $f_{10} = 10^{-10} + 10^{-10} + 10^{-10} + 10^{-10} + 10^{-10} + 10^{-10} + 10^{-10} + 10^{-10} + 10^{-10} + 10^{-10} + 10^{-10} + 10^{-10} + 10^{-10} + 10^{-10} + 10^{-10} + 10^{-10} + 10^{-10} + 10^{-10} + 10^{-10} + 10^{-10} + 10^{-10} + 10^{-10} + 10^{-10} + 10^{-10} + 10^{-10} + 10^{-10} + 10^{-10} + 10^{-10} + 10^{-10} + 10^{-10} + 10^{-10} + 10^{-10} + 10^{-10} + 10^{-10} + 10^{-10} + 10^{-10} + 10^{-10} + 10^{-10} + 10^{-10} + 10^{-10} + 10^{-10} + 10^{-10} + 10^{-10} + 10^{-10} + 10^{-10} + 10^{-10} + 10^{-10} + 10^{-10} + 10^{-10} + 10^{-10} + 10^{-10} + 10^{-10} + 10^{-10} + 10^{-10} + 10^{-10} + 10^{-10} + 10^{-10} + 10^{-10} + 10^{-10} + 10^{-10} + 10^{-10} + 10^{-10} + 10^{-10} + 10^{-10} + 10^{-10} + 10^{-10} + 10^{-10} + 10^{-10} + 10^{-10} + 10^{-10} + 10^{-10} + 10^{-10} + 10^{-10} + 10^{-10} + 10^{-10} + 10^{-10} + 10^{-10} + 10^{-10} + 10^{-10} + 10^{-10} + 10^{-10} + 10^{-10} + 10^{-10} + 10^{-10} + 10^{-10} + 10^{-10} + 10^{-10} + 10^{-10} + 10^{-10} + 10^{-10} + 10^{-10} + 10^{-10} + 10^{-10} + 10^{-10} + 10^{-10} + 10^{-10} + 10^{-10} + 10^{-10} + 10^{-10} + 10^{-10} + 10^{-10} + 10^{-10} + 10^{-10} + 10^{-10} + 10^{-10} + 10^{-10} + 10^{-10} + 10^{-10} + 10^{-10} + 10^{-10} + 10^{-10} + 10^{-10} + 10^{-10} + 10^{-10} + 10^{-10} + 10^{-10} + 10^{-10} + 10^{-10} + 10^{-10} + 10^{-10} + 10^{-10} + 10^{-10} + 10^{-10} + 10^{-10} + 10^{-10} + 10^{-10} + 10^{-10} + 10^{-10} + 10^{-10} + 10^{-10} + 10^{-10} + 10^{-10} + 10^{-10} + 10^{-10} + 10^{-10} + 10^{-10} + 10^{-10} + 10^{-10} + 10^{-10} + 10^{-10} + 10^{-10} + 10^{-10} + 10^{-10} + 10^{-10} + 10^{-10} + 10^{-10} + 10^{-10} + 10^{-10} + 10^{-10} + 10^{-10} + 10^{-10}$

20 for SO2) may be less than 30 s. Take 2019-3-29A for example, part of original measurement data is as bellow:

| Time     | SO 2 (ppm) | CO 2 (ppm) | Time     | SO 2 (ppm) | CO 2 (ppm) |
|----------|-----------------------|-----------------------|----------|-----------------------|-----------------------|
| 10:38:09 | 0                     | 3373                  | 10:38:25 | 0.25                  | 2504                  |
| 10:38:10 | 0                     | 3342                  | 10:38:26 | 0.28                  | 2636                  |
| 10:38:11 | 0                     | 3272                  | 10:38:27 | 0.44                  | 2813                  |
| 10:38:12 | 0.18                  | 3176                  | 10:38:28 | 0.75                  | 3287                  |
| 10:38:13 | 0.06                  | 3063                  | 10:38:29 | 1.43                  | 3558                  |
| 10:38:14 | 0.17                  | 2943                  | 10:38:30 | <mark>1.63</mark>     | <mark>3822</mark>     |
| 10:38:15 | 0.37                  | 2775                  | 10:38:31 | 1.45                  | 4074                  |
| 10:38:16 | 0.19                  | 2712                  | 10:38:32 | 1.21                  | 4290                  |
| 10:38:17 | 0.37                  | 2675                  | 10:38:33 | 0.72                  | 4417                  |
| 10:38:18 | 0.16                  | 2646                  | 10:38:34 | 0.34                  | 4622                  |
| 10:38:19 | 0.1                   | 2619                  | 10:38:35 | 0.55                  | 4662                  |
| 10:38:20 | 0.29                  | 2584                  | 10:38:36 | <mark>0.61</mark>     | <mark>4671</mark>     |
| 10:38:21 | 0.46                  | 2537                  | 10:38:37 | 0.11                  | 4660                  |
| 10:38:22 | 0.51                  | 2423                  | 10:38:38 | 0.2                   | 4622                  |
| 10:38:23 | 0.41                  | 2409                  | 10:38:39 | 0                     | 4560                  |
| 10:38:24 | 0.44                  | 2426                  | 10:38:40 | 0                     | 4469                  |

Table 1 part of original measurement data of 2019-3-29A

"to numerically degrade the SO2 time series to correspond to the time constant of the CO2 instrument". A related question is, whether you have used exactly the same time for the readings of the maximum values of SO2 and CO2, although there is an obvious time shift between the two.

25

This is one approach that can be taken. Assume that the data in Table 1 are average data within 10s and can be used as the selection values for the FSC. There are three ways to select peak values:

Case 1: 1.63 for SO2, 3822 for CO2; The result value of FSC is relatively large.

Case 2: 0.61 for SO2, 4671 for CO2; The result value of FSC is relatively small.

Case 3: 1.63 for SO2, 4671 for CO2; The result value of FSC is relatively moderate.

What we did before was the option 1. The reason for this is that some of the sulfur is not converted to  $SO_2$  (possibly  $SO_3$  or  $SO_4$ ). Choosing a larger result value may be closer to the true value.

In our first-generation pod (Zhou et al., 2019). Overall, despite the use of option 1, there was a high incidence of low estimates as shown in Figure 1.

35

Figure 1. Comparison between the true values of FSC (x-axis) against the estimated values of FSC (y-axis) of 23 times measurement.

Therefore, the first-generation pod is suitable for adopting option 1 to obtain more accurate FSC.

It should be noted that the sensors of the first- and second-generation pod were provided by different manufacturer.

40 In our second-generation pod (this research), I also have done comparisons as listed below in Table 2. The deviation of results obtained in option 1 and 3 were similar on the whole. My initial choice is option 1, but option 3 is also OK.

| Table 2 Comparison of option 1 and option 3 |                      |                      |                   |  |  |  |
|---------------------------------------------|----------------------|----------------------|-------------------|--|--|--|
| ID                                          | Estimated FSC of     | Estimated FSC of     | True value of FSC |  |  |  |
|                                             | option 3 (deviation) | option 1 (deviation) |                   |  |  |  |
| 2019-3-18A                                  | 0.207 (-0.015)       | 0.217 (-0.005)       | 0.222             |  |  |  |
| 2019-3-22A                                  | 0.062 (-0.037)       | 0.069 (-0.030)       | 0.099             |  |  |  |
| 2019-3-22B                                  | 0.046 (0.004)        | 0.046 (0.004)        | 0.042             |  |  |  |
| 2019-3-29A                                  | 0.051 (0.016)        | 0.057 (0.022)        | 0.035             |  |  |  |
| 2019-4-1A                                   | 0.064 (-0.015)       | 0.090 (0.011)        | 0.079             |  |  |  |
| 2019-4-3A                                   | < 0.020              | < 0.020              | 0.013             |  |  |  |
| 2019-4-3B                                   | 0.052 (-0.040)       | 0.057 (-0.035)       | 0.092             |  |  |  |
| 2019-4-12A                                  | < 0.020              | < 0.020              | 0.004             |  |  |  |
| 2019-4-12B                                  | 0.080 (0)            | 0.092 (0.012)        | 0.080             |  |  |  |
| 2019-4-15A                                  | 0.035 (-0.009)       | 0.053 (0.009)        | 0.044             |  |  |  |
| 2019-4-15B                                  | 0.154 (-0.014)       | 0.167 (-0.001)       | 0.168             |  |  |  |

45 Based on the above analysis and combined with your suggestions, I think option 3 is more appropriate. Because option 1 is, after all, an empirical choice, it is difficult to describe quantitatively.

But please note that for the option 3 is chosen. All the FSC estimated result in Tables 2-4, Figure 6, and some description in the original manuscript have been updated.

Meanwhile, I added the following discussion.

50 Meanwhile, the occurrence times of the peak  $SO_2$  and  $CO_2$  values sometimes have a small deviation that usually corresponds to a few seconds. This is due to two different sensor response times, which leads to three different options for selecting the peak values: 1) the time point of the peak  $SO_2$  value with the  $CO_2$  value at the same time; 2) the time point of the peak  $CO_2$  value with the  $SO_2$  value at the same time; 3) the peak  $SO_2$  and  $CO_2$  values at different time points. Option 3 was selected in this research.

- 60 One of the reviewers pointed out several important issues which could affect the accuracy of the measurements, including cross sensitivity to NO2, the impact of large temperature changes in the exhaust plume and the issues of water vapour and particle contamination of the instruments. Please add a short discussion of these points to the section on uncertainties.
- 65 Ok, I have added a short discussion as bellow:

Zhou, F., Pan, S., Chen, W., Ni, X., and An, B.: Monitoring of compliance with fuel sulfur content regulations through unmanned aerial vehicle (UAV) measurements of ship emissions, Atmos. Meas. Tech., 12, 6113–6124, https://doi.org/10.5194/amt-12-6113-2019, 2019.

To make it lightweight and convenient, the second-generation pod was only equipped with  $SO_2$  and  $CO_2$  sensors and a simple filter. We did not account for the interference that some factors might have caused, including that due to 1) the cross-sensitivity of the  $SO_2$  sensor to  $NO_2$ , 2) the impact of a large temperature change in the exhaust plume, and 3) water vapor and/or particle contamination of the instruments.

**In Figure 5, highlight the points used for the FSC computation.**

Ok, I have added.

70

75 In Figure 6, please indicate by a dotted vertical line the FSC limit of 0.5%. Ok, I have added.

Quantification of poor measurements: Why don't you use the correlation between SO2 and CO2 time series to determine the quality of a measurement? In my opinion, this should be a good first guess of which measurements

80 to use and which to discard.

Yes, it is indeed a good first guess of which measurements to use and which to discard. I have added the discussion in the manuscript.

Meanwhile, the correlation between the SO2 and CO2 time series is a key factor in judging quality. Assuming that the

gas is completely mixed, the variation trend of the  $SO_2$  and  $CO_2$  measurements should be the same (although there may be some deviation because the corresponding time of the sensor was not consistent) and can be identified in the peak area.

**Comment from the Referee #2:**

90 The authors have improved their study significantly, following the suggestions by the reviewers. I recommend it for publication in AMT.

One short comment: The mixed use of UAV and UAS is sometimes a bit confusing and also not always used in a proper way. The authors should think to skip out one of these very similar acronyms.

OK, after statistics, the rating rate of UAV use was significantly higher. I modified the text to use only the UAV, and the UAS was replace of UAV system.

**Other modifications:**

95

On the basis of the research about ship emission monitoring, we have established a provincial-level research center:

Shanghai Engineering Research Center of Ship Exhaust Intelligent Monitoring. Because the name was not determined

100 when this manuscript was submitted, I did not add this institution name. Now, the institution has been formally established.

I apply to add this in the authors' institution information, if the rules allow it.

Finally, thank you for your suggestions, which not only improves the quality of the manuscript, but also makes me more aware of what I need to research in the future.

**Monitoring compliance with fuel sulfur content regulations of sailing ships by unmanned aerial vehicle (UAV) measurements of ship emissions in open water**

Fan Zhou1,2, Liwei Hou2Hou3, Rui Zhong1,2, Wei Chen3Chen4, Xunpeng Ni3Ni4, Shengda Pan1,2, Ming Zhao1,42,5, Bowen An1,2

[revised manuscript text omitted]